# In situ architecture of Opa1-dependent mitochondrial cristae remodeling

Michelle Y Fry[1,2,14], Paula P Navarro[1,2,11,14], Pusparanee Hakim [1], Virly Y Ananda[1], Xingping Qin [3,4,5], Juan C Landoni [6], Sneha Rath[1], Zintis Inde [3,4,5], Camila Makhlouta Lugo [1], Bridget E Luce[1], Yifan Ge [1,2,12], Julie L McDonald [1,13], Ilzat Ali[1,2], Leillani L Ha[7,8], Benjamin P Kleinstiver[7,8,9], David C Chan [10], Kristopher A Sarosiek[3,4,5] & Luke H Chao [1,2 ✉]

## Abstract

Cristae membrane state plays a central role in regulating mitochondrial function and cellular metabolism. The protein Optic atrophy 1 (Opa1) is an important crista remodeler that exists as two forms in the mitochondrion, a membrane-anchored long form (l-Opa1) and a processed short form (s-Opa1). The mechanisms for how Opa1 influences cristae shape have remained unclear due to lack of native three-dimensional views of cristae. We perform in situ cryo-electron tomography of cryo-focused ion beam milled mouse embryonic fibroblasts with defined Opa1 states to understand how each form of Opa1 influences cristae architecture. In our tomograms, we observe a variety of cristae shapes with distinct trends dependent on s-Opa1:l-Opa1 balance. Increased l-Opa1 levels promote cristae stacking and elongated mitochondria, while increased s-Opa1 levels correlated with irregular cristae packing and round mitochondria shape. Functional assays indicate a role for l-Opa1 in wild-type apoptotic and calcium handling responses, and show a compromised respiratory function under Opa1 imbalance. In summary, we provide three-dimensional visualization of cristae architecture to reveal relationships between mitochondrial ultrastructure and cellular function dependent on Opa1-mediated membrane remodeling.

**Keywords** Cristae Remodeling; Mitochondrial Biology; Cryo-Electron Tomography; Cryo-Focused Ion Beam Milling
**Subject Categories** Organelles; Structural Biology

## Introduction

Mitochondria play essential roles in energy production, metabolism and signaling, which drive diverse biological functions and processes of the cell (Chandel, 2014; Spinelli and Haigis, 2018; Picard and Shirihai, 2022). The organelle undergoes membrane remodeling during homeostatic steady-state conditions through fusion and fission to generate a dynamic and responsive reticulum (Twig et al, 2008; Mishra et al, 2014; Mishra and Chan, 2016). Mitochondrial ultrastructure is defined by a double-membrane architecture with an outer (OMM) and inner mitochondrial membrane (IMM). The IMM can be divided into three subregions, the cristae folds, the inner boundary membrane (IBM) and the cristae junctions (CJ), which separates cristae from the IBM. Cristae are enriched with electron transport chain (ETC) proteins, making them a hub for energy production (Schägger and Pfeiffer, 2000). In metazoans, cristae are mostly lamellar or tubular (Hashimi, 2019) and undergo dramatic rearrangements in response to stress and during initiation of apoptotic cell death (Frey and Mannella, 2000; Mannella, 2006; Zick et al, 2009). Despite longstanding interest in mitochondrial morphogenesis and major advances in understanding of mitochondrial structure and function (Friedman and Nunnari, 2014), regulation of cristae remodeling remains poorly understood.

Relevant proteins for cristae remodeling include ATP synthase (Blum et al, 2019), optic atrophy protein 1 (Opa1) and the mitochondrial contact site and cristae organizing system (MICOS) (Friedman and Nunnari, 2014; Laan et al, 2016). Opa1 is responsible for fusing the IMM during mitochondrial fusion and is also implicated in cristae remodeling. Inner-membrane proteases, such as Oma1, Yme1L, and ParI, process Opa1 to generate a "soluble" form that lacks the N-terminal transmembrane (TM) segment (s-Opa1) (Song et al, 2007; Anand et al, 2014). This processing results in two forms of Opa1 in the mitochondria: the

[1]Department of Molecular Biology, Massachusetts General Hospital, Boston, MA, USA. [2]Department of Genetics, Blavatnik Institute, Harvard Medical School, Boston, MA, USA. [3]John B. Little Center for Radiation Sciences, Harvard T.H. Chan School of Public Health, Boston, MA, USA. [4]Molecular and Integrative Physiological Sciences (MIPS) Program, Harvard T.H. Chan School of Public Health, Boston, MA, USA. [5]Lab of Systems Pharmacology, Harvard Program in Therapeutic Science, Department of Systems Biology, Harvard Medical School, Boston, MA, USA. [6]Institute of Physics, École Polytechnique Fédérale de Lausanne (EPFL), Lausanne, Switzerland. [7]Center for Genomic Medicine, Massachusetts General Hospital, Boston, MA, USA. [8]Department of Pathology, Massachusetts General Hospital, Boston, MA, USA. [9]Department of Pathology, Harvard Medical School, Boston, MA, USA. [10]Division of Biology and Biological Engineering, California Institute of Technology, Pasadena, CA, USA. [11]Present address: Department of Microbiology, Blavatnik Institute, Harvard Medical School, Boston, MA, USA. [12]Present address: Interdisciplinary Research Center of Biology and Chemistry, Shanghai Institute of Organic Chemistry, Chinese Academy of Science, Shanghai, China. [13]Present address: Department of Biology, Massachusetts Institute of Technology, Cambridge, MA, USA. [14]These authors contributed equally: Michelle Y Fry, Paula P Navarro. ✉E-mail: chao@molbio.mgh.harvard.edu

soluble short form (s-Opa1) and the unprocessed, full N-terminal TM anchored long form (l-Opa1). We previously showed that stoichiometric levels of both l- and s-Opa1 are required for fast and efficient membrane fusion in vitro (Ge et al, 2020); however, in situ ultrastructural understanding of how cristae shape is maintained by each Opa1 form is lacking.

Previous work used transmission electron microscopy (TEM) imaging of chemically fixed and heavy metal-stained cells to investigate the role of Opa1 in cristae shape. Knocking-down Opa1 induced more disorganized, globular and hyper-convex cristae, whereas over expression of Opa1 resulted in narrower cristae and CJs (Frezza et al, 2006). Initial steps of apoptosis involve release of cytochrome c from the cristae lumen, where ~85% of the cellular cytochrome c is stored (Scorrano et al, 2002). Expression of wild-type (WT) Opa1 was reported to protect against apoptosis by restricting cytochrome c release, whereas the loss of Opa1 was reported to result in mitochondria fragmentation and enhanced cytochrome c release (Scorrano et al, 2002). Addition of pro-apoptotic peptides resulted in wider CJs observed in TEM images. These observations suggest the organelle's inner-membrane functional state may be sensitive to the levels of Opa1 present, and the membrane architectures supported by Opa1.

To understand the role of s- and l-Opa1 in cristae architecture and remodeling, we investigated the in situ state of mitochondrial membranes in mouse embryonic fibroblasts (MEF) with different levels of s- and l-Opa1, by applying cryo-electron tomography (cryo-ET) on cryo-focused ion beam (cryo-FIB) milled MEF. We used defined cell lines that predominantly contained either l- or s-Opa1 to characterize how mitochondrial membrane organization and shape depend on the expression levels and form of Opa1. Here, we present an extensive characterization of the 3D morphological properties of mitochondrial cristae membranes. We observed that l-Opa1 contributes to crista stacking, longer cristae, a reduction of globular cristae, an absence of tubular cristae and the maintenance of cristae junction widths. We found that the presence of s-Opa1 correlates with tubular cristae, wider cristae junctions, narrower cristae, and irregular cristae packing. Using BH3 profiling, we observed WT-like apoptotic responses only in the cells with l-Opa1. We also found WT-like calcium handling responses in cells expressing mainly l-Opa1 and that both forms of Opa1 are important for normal respiratory function. Our work reveals roles for s- and l-Opa1 beyond mitochondrial fusion and demonstrates that both forms of Opa1 play distinct roles in cristae shape maintenance important for mitochondrial function.

# Results

## In situ morphology of mitochondria with different Opa1 processing

We generated cryo-electron tomograms of mitochondria from cryo-FIB milled MEF cell lines that differed in the expression levels and processed states of Opa1, to investigate mitochondrial cristae ultrastructure (Appendix Fig. S1). In this study, we used five MEF cell lines: (i) wild-type (WT) cells, (ii) cells stably overexpressing Opa1 (Opa1-OE), (iii) a Δexon5b CRISPR MEF line with an oma1-/- background, which restricts Opa1 cleavage and results in the

presence of mostly l-Opa1 (referred to as l-Opa1* in this work), (iv) an Opa1 knock-out (Opa1-KO) line stably expressing the Opa1 isoform 5, which is robustly processed and results in the presence of s-Opa1 (referred to as s-Opa1*) and (v) Opa1-KO cells (Mishra et al, 2014; Wang et al, 2021; Appendix Fig. S1). While previous studies have characterized mitochondrial ultrastructure, these samples were subjected to chemical fixation and heavy metal staining, which are recognized to perturb cellular membrane state and limit resolution (Bäuerlein and Baumeister, 2021). Thus, we vitrified MEF cell lines to characterize mitochondrial membrane architecture under native conditions. To prepare samples for cryo-ET imaging, we generated ~200-350 nm thick lamellae by cryo-FIB milling (Rigort et al, 2012; Mahamid et al, 2016), following a previously established imaging pipeline (Navarro et al, 2022). A total of 100 tilt-series (WT = 33, Opa1-OE = 7, l-Opa1* = 21, s-Opa1* = 28 and Opa1-KO = 11) were acquired, aligned and three-dimensionally (3D) reconstructed into tomograms (Table EV1). Cryo-electron tomograms were denoised using Topaz-Denoise to improve mitochondrial membrane visualization (Bepler et al, 2020). It is important to note that cryo-electron tomograms generated by cryo-FIB milling cover a section of the mitochondria contained within the thickness of the lamella, thus, all mitochondria were partially imaged in the Z-axis. In some cases, mitochondria are only partially visible in the XY plane due to the trade-off between resolution and field of view (Navarro, 2022; Appendix Fig. S2A). Densities corresponding to the IMM and OMM were 3D segmented in yellow and green, respectively (Fig. 1A, Movies EV1–5).

The generated tomograms provide rich 3D information on mitochondrial morphology. By visual morphological analysis, we classified mitochondria into ellipsoidal, round, partial (when partially imaged in the XY plane) or polygon shape categories. The majority of mitochondria in all cell lines are ellipsoidal, except for s-Opa1* cells, where round mitochondria dominate (Fig. 1B), correlating with a smaller apparent mitochondrial area (Fig. 1C). Mitochondria in WT cells are larger in area $(0.34 \pm 0.03 \ \mu m^2)$ than in other cell lines; with smaller mitochondria in l-Opa1* cells $(0.2 \pm 0.02 \ \mu m^2)$ and significantly smaller mitochondria in Opa1-OE $(0.14 \pm 0.02 \ \mu m^2)$ and s-Opa1* cells $(0.18 \pm 0.02 \ \mu m^2)$ (Fig. 1C). We measured the mitochondrial volume and the volumes of each subcompartment: the matrix, inner membrane space (IMS) and cristae lumen (CL) (Fig. EV1). Our analysis shows that imbalance of l- and s-Opa1 levels results in larger CL volume (Fig. EV1C–F).

At first glance, we observed differences in matrix contrast between mitochondria from different cell lines in our cryo-ET data. To investigate this further, we quantified the gray scale levels in normalized summed projected images from cryo-electron tomograms (Fig. EV1G). We observed denser mitochondrial matrices in Opa1-OE, l-Opa1*, and Opa1-KO mitochondria. For the Opa1-OE mitochondria, the darker measured gray scale value can be attributed to the presence of electron dense deposits within the mitochondrial matrix, likely to be calcium phosphate deposits (Wolf et al, 2017; Strubbe-Rivera et al, 2021; Figs. 1A and S2A white arrowheads, Fig. EV1H).

We analyzed mitochondrial cristae density (number of cristae/ $\mu m^2$) in the cryo-electron tomograms. No significant difference was observed in the cristae density between mitochondria in WT and Opa1-KO cells (Fig. EV2A). All other conditions (Opa1-OE,

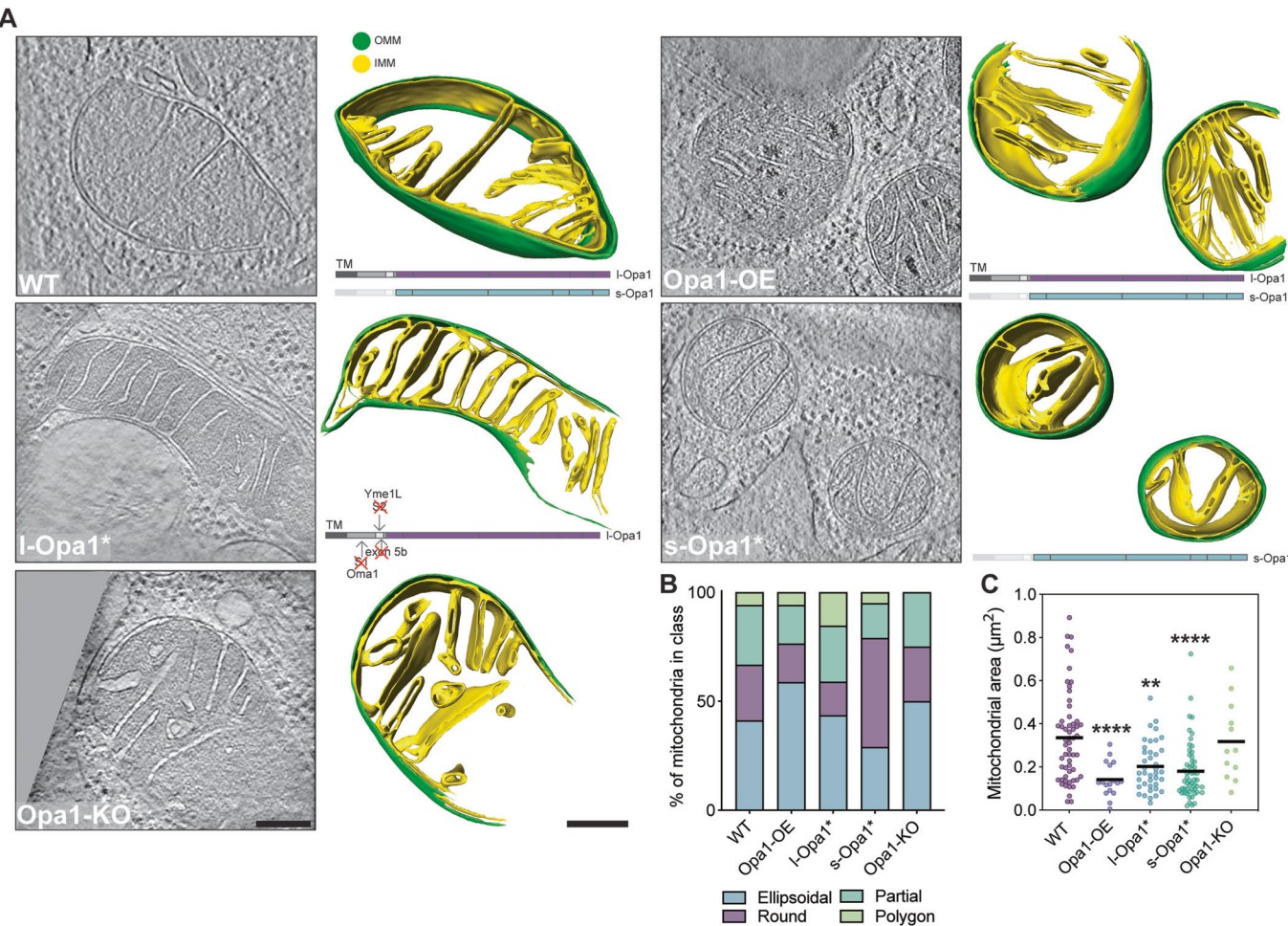

**Figure 1.  In situ mitochondrial membrane morphology is influenced by Opa1 processing.**

Mitochondria with distinguishable inner mitochondrial membrane (IMM) and outer mitochondrial membrane (OMM) visualized by cryo-ET. (A) (Right) Summed, projected central slices of cryo-electron tomograms visualizing representative mitochondria in indicated MEF cell lines. (Left) Three-dimensional (3D) rendering of segmented membranes with mitochondria shown across Z slices. Green and yellow surfaces indicate OMM and IMM, respectively. (Bottom right) Schematic of Opa1 forms present in respective cell lines. Scale bar = 200 nm. (B) Graph bar representing the relative proportion of different mitochondrial shapes observed. *N* refers to number of mitochondria: wild-type = 57, Opa1-OE = 17, l-Opa1* = 39, s-Opa1* = 55, Opa1-KO = 12. (C) Plot of mitochondria size (µm²) observed in cryo-electron tomograms in MEF lines. *N* refers to number of mitochondria: wild-type = 57, Opa1-OE = 17, l-Opa1* = 39, s-Opa1* = 55, Opa1-KO = 12. Data Information: Scatter plots show data distribution, the mean is marked by a bold black line. Significance of difference is tested relative to wild type using Mann–Whitney test; **p < 0.005, ****p < 0.0001.  Source data are available online for this figure.

l-Opa1* and s-Opa1*) resulted in a significant increase in cristae density compared to WT. Such variation was not reflected in the total number of cristae per mitochondria (Fig. EV2B). This can be explained by the reduction in average mitochondria size (Fig. 1C). These observations indicate that altered Opa1 levels or processing can impact the steady-state cristae number in a given mitochondrial volume.

We next characterized how Opa1 processing influences cristae organization. In l-Opa1* cells, we observed organized and parallel oriented cristae, whereas cristae from s-Opa1* cells do not exhibit such a pattern or organization and frequently cross over one another along the z-axis (Figs. 1A, S2A, and EV2C). Additionally, we observed a subpopulation of mitochondria displaying a stacking cristae phenotype, which we defined as three or more lamellar cristae running in parallel to one another into the mitochondrial

matrix throughout the tomogram (Fig. EV2C). This phenotype was observed at similar levels in WT (23.53%) and s-Opa1* cells (27.27%), but slightly more frequently in Opa1-KO (33.33%) cells. Opa1 overexpression or inhibition of Opa1 processing (l-Opa1*) resulted in a dramatic increase (>50%) of mitochondria with stacking cristae (Fig. EV2D). These results suggest that homeostatic levels of Opa1 processing maintain WT levels of cristae stacking.

We also used live-cell fluorescence microscopy (FM) and immunofluorescence to assess whole cell mitochondrial network morphology. Consistent with previous FM reports, we observed an almost equal mixture of short and long tubular mitochondrial networks in WT cells and a dramatically fragmented network in Opa1-KO cells (Song et al, 2007; Patten et al, 2014; Bocca et al, 2018; Fig. EV3). While fragmented mitochondrial networks were observed in s-Opa1* cells, more elongated tubular networks were

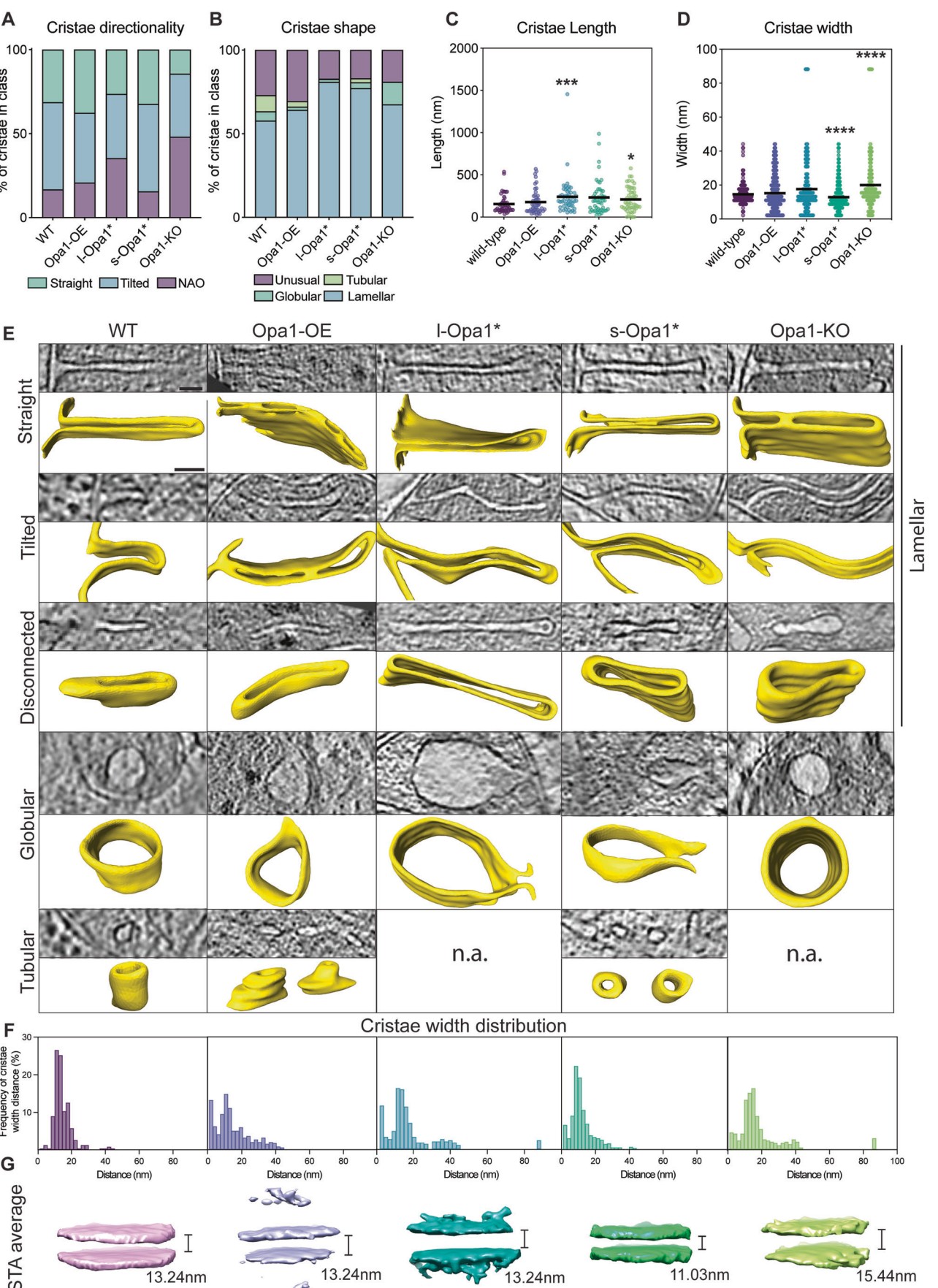

**Figure 2. In situ crista ultrastructure.**

(A) Graph bars showing the proportions of straight, tilted, and no attachment observed (NAO) crista observed in indicated MEF lines. *N* refers to the number of cristae analyzed: WT = 131, Opa1-OE = 166, l-Opa1* = 380, s-Opa1* = 495, Opa1-KO = 112. (B) Graph bars showing the proportions of lamellar, tubular, globular and unusual crista observed in indicated MEF lines. N refers to the number of cristae analyzed: WT = 131, Opa1-OE = 166, l-Opa1* = 380, s-Opa1* = 495, Opa1-KO = 112. (C) Measured cristae length across cell lines. *N* = 50 for all cell lines. (D) Measured cristae width across cell lines. *N* refers to the number of cristae subvolumes: WT = 222, Opa1-OE = 430, l-Opa1* = 323, s-Opa1* = 653, Opa1-KO = 243. (E) (Top rows) Computational slices of straight, tilted, disconnected, globular and tubular crista across cell lines and the corresponding 3D renderings (bottom rows) from cryo-electron tomograms (n.a. = not applicable). Scale bar = 50 nm. (F) Histograms of crista widths across cell conditions (see Methods). *N* refers to the number of cristae subvolumes: WT = 222, Opa1-OE = 430, l-Opa1* = 323, s-Opa1* = 653, Opa1-KO = 243. (G) Subtomogram averages of mitochondrial cristae membranes with the average width indicated. Data information: Scatter plots show data distribution, the mean is marked by a bold black line. Removal of outliers did not change statistical test results (see source data). Significance of difference is tested relative to WT using Mann–Whitney test; *$p < 0.05$, ***$p < 0.001$, ****$p < 0.0001$. For (D), mean ± SD in WT = 14.5 ± 5.8; Opa1-OE = 15.2 ± 10.5; l-Opa1* = 17.6 ± 15.4; s-Opa1* = 12.9 ± 7.9; Opa1-KO = 20 ± 15.8. Source data are available online for this figure.

observed in l-Opa1* cells correlating with the increased number of round and polygon mitochondria respectively found by cryo-ET (Fig. 1A,B). Opa1-OE cells showed longer tubular networks, compared to WT (Fig. EV3B). These data suggest healthy fusion activity in WT and l-Opa1* cells and corroborate our cryo-ET observations (Fig. EV3, Movie EV6).

## In situ cristae ultrastructure

To address questions related to the specific roles for Opa1 forms in cristae morphology, we sought to characterize the dependence of cristae architecture on the form of Opa1 present. Based on their direction relative to the OMM, we quantified cristae as straight or tilted. For some cristae, no connection to the IMM was captured in the tomogram and thus classified as no attachment observed (NAO) (Fig. 2A). This is a consequence of all mitochondria being partially imaged in the z-axis, as mentioned above. In Opa1-KO cells fewer cristae were straight compared to WT (Fig. 2A). The presence of either form of Opa1 restores the proportion of straight cristae to WT-like levels, while overexpression of Opa1 results in more straight cristae than WT.

Cristae were then classified based on 3D-shape into canonical lamellar, globular, and tubular categories (Harner et al, 2016). Across all cell lines, lamellar cristae dominate (Fig. 2B). Interestingly, the proportion of lamellar cristae increases in l-Opa1* (81%) and s-Opa1* (77%) cells. Tubular cristae are not observed in l-Opa1* and Opa1-KO cells and are reduced in Opa1-OE (3.24%) and s-Opa1* (2.3%) cells compared to WT (9.6%). Globular cristae are present in all cell lines albeit to a lesser extent when Opa1 expression levels are altered but increased in the absence of Opa1 (Fig. 2B). The proportion of unusually shaped cristae is similar among WT (27%) and Opa1-OE (30.52%), but reduced in l-Opa1* (17%), s-Opa1* (16.9%) and Opa1-KO (18.9%) cell lines (Fig. 2B, Appendix Fig. S2A).

Unusual cristae were further subclassified into eight defined categories: (i) loop, where cristae curve and connect to IBM via two CJs; (ii) split, where cristae branch into two or more cristae; (iii) straight-across, when cristae are perpendicular and connect to the IBM via two CJs; (iv) amorphous, when cristae display a nebulous morphology; (v) ring, where cristae are circular; (vi) pinched, where cristae show areas where membranes touch; (vii) zipped, when cristae have regions where both membranes merge until both cannot be distinguished; and (viii) vesicular, where material is observed within the cristae (Fig. EV4). Of these categories only five were observed in WT mitochondria—ring, loop, straight-across,

amorphous, and split. The loop phenotype is dominant in WT unusual cristae, but also present in all cell lines (Fig. EV4A). Pinched and zipped cristae are absent in WT cells but predominant in Opa1-OE cells. Zipped cristae were observed in l-Opa1* cells and only once in s-Opa1* cells, suggesting l-Opa1 may play a role in bridging the two membranes of the cristae for long stretches. In the absence of Opa1, no split or straight across cristae were observed; instead, more cristae fall into the amorphous and vesicular categories. An increase in amorphous cristae were also observed in l-Opa1* and s-Opa1* cells. All cristae shapes were observed in s-Opa1* cells, with roughly a quarter of the unusual cristae falling into the pinched category. Compared to WT, a similar proportion of unusual cristae in s-Opa1* mitochondria fall into the split category. Ring shaped cristae are rare and were not observed in l-Opa1* cells.

To determine the dependence of cristae length and width on the form of Opa1 present, we measured in 3D the length and width of fifty cristae per cell line (Appendix Fig. S3A). Knocking out Opa1 results in significantly longer cristae (Fig. 2C). This increase in cristae length was not observed in Opa1-OE cells, but was observed in both s-Opa1* and l-Opa1* cells. In s-Opa1* mitochondria, there is a broad distribution of cristae lengths, with a considerable number of longer cristae than WT, which was also observed in Opa1-KO mitochondria. Overexpression of Opa1 does not affect cristae width, but the absence of Opa1 correlates with wider cristae, consistent with a larger proportion of observed globular cristae (Fig. 2D). Cristae widths are restored to WT values in l-Opa1* mitochondria, but in s-Opa1* mitochondria cristae are significantly narrower. Though Opa1-OE and l-Opa1* mitochondria have similar average cristae widths to WT mitochondria, a larger variation in cristae widths was observed, with many values falling into 0-5 nm and 9-14 nm ranges (Fig. 2F,G). This suggests that a significant proportion of cristae in Opa1-OE and l-Opa1* mitochondria vary in width, correlating with an increase in the number of zipped or pinched cristae in these cell lines (Fig. 2E). Even though cristae width histograms from s-Opa1* and WT mitochondria exhibit a single narrow peak, s-Opa1* mitochondria have tighter cristae (6–15 and 9–16 nm, respectively) (Fig. 2E–G). In Opa1-KO mitochondria, the cristae width distribution peak ranges from 10 to 17 nm, with an outlier representing a population of extremely wide cristae also seen in l-Opa1* mitochondria, particularly a subclass of globular cristae (Fig. 2B,D–G). These observations suggest the uniformity of cristae widths observed in WT mitochondria is disrupted by changes in Opa1 expression levels and suppression of Opa1 processing.

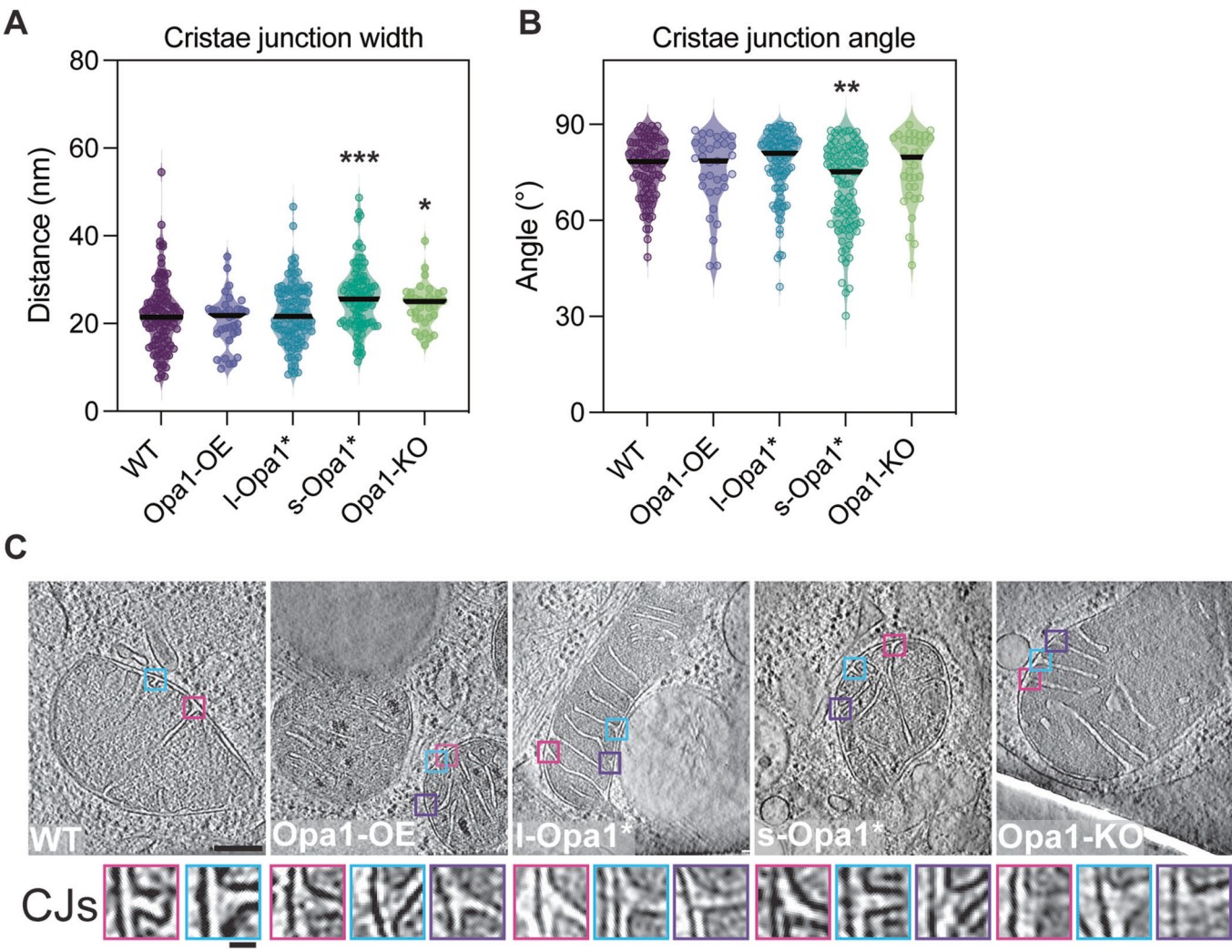

**Figure 3. Quantification of cristae junction (CJ) properties.**

(A) Plot of measured cristae junction width across cell lines. (see Appendix Fig. S3C,D and "Methods" section for measurement methods). *N* refers to number of cristae analyzed: WT = 103, Opa1-OE = 33, l-Opa1* = 107, s-Opa1* = 92, Opa1-KO = 34. (B) Plot of measured cristae junction angle across cell lines (see Appendix Fig. S3C,D and "Methods" section for measurement methods). *N* refers to number of cristae analyzed: WT = 103, Opa1-OE = 33, l-Opa1* = 107, s-Opa1* = 92, Opa1-KO = 34. (C) (Top) Summed, projected central slices of cryo-electron tomograms of representative mitochondria analyzed in (A, B) with magnified cristae junction (bottom insets). WT, Opa1-OE, l-Opa1* and Opa1-KO are the same tomograms from Fig. 1A. Scale bar = 100 nm. Inset scale bar = 25 nm. Data information: Scatter plots show data distribution, the mean is marked by a bold black line. Significance of difference is tested relative to wild type using Mann–Whitney test; *$p < 0.05$, **$p < 0.01$, ***$p < 0.001$. Source data are available online for this figure.

## Cristae junction morphology by cryo-ET

To gain insight into the role of each Opa1 form at the cristae junction (CJ), we measured and analyzed CJs from summed projected central slices of tomograms for all cell lines. CJs were defined as the site where the crista membrane joins the boundary region of the IMM (Appendix Fig. S3C). Compared to WT mitochondria, we observed wider CJs in Opa1-KO mitochondria, consistent with previous TEM data (Scorrano et al, 2002) and unaltered CJ widths in Opa1-OE mitochondria (Fig. 3A). Intriguingly, we found that CJs are also significantly wider in s-Opa1* mitochondria compared to WT mitochondria (Fig. 3A). We did not, however, find any significant difference in CJ widths in l-Opa1* mitochondria. These measurements indicate under

steady-state conditions complete processing of Opa1 results in larger CJ widths.

In our initial inspection of our tomograms, we noticed that a greater proportion of cristae in s-Opa1* mitochondria are tilted, i.e., where the crista length is not perpendicular to the IBM (Fig. 2A). To further investigate if this tilt occurs at the beginning of the crista or along the length of the cristae, we quantified the angle at the CJ (Appendix Fig. S3D). While most cristae are orthogonal to the OMM, we observed a wide range of CJ angles in s-Opa1* mitochondria. Overall, the CJ angles in s-Opa1* cristae are less perpendicular compared to WT CJs, demonstrating that the tilted cristae observed in these cells occurs at the CJ (Figs. 2A and 3B,C).

We also found that some cristae have two or more CJs. Most l-Opa1* mitochondria (77%) have at least one crista with multiple

junctions, which is considerably higher than WT (35%), Opa1-OE (30%), s-Opa1* (27%), and Opa1-KO (25%) mitochondria (Fig. EV5A). These multijunction cristae were further classified as straight or loop. In WT cells, there is an equivalent number of straight and loop cristae with more than one CJ and in Opa1-KO mitochondria, the ratio shifts to favor the loop class (Fig. EV5B). In Opa1-OE and l-Opa1* mitochondria, the majority of multijunction cristae fall into the straight category, correlating with a greater proportion of straight-across cristae shapes observed in l-Opa1* mitochondria (Figs. 3B and EV5B).

## Functional consequences of Opa1 state

We next examined the consequences of Opa1 levels and processing on mitochondrial function. First, we sought to uncover potential effects of Opa1 perturbation on the intrinsic apoptotic pathway. We used BH3 profiling to compare mitochondrial responses to pro-apoptotic stimuli across the cell lines. BH3 profiling quantifies the percentage of cells that release cytochrome c from their mitochondria, the key commitment step in mitochondrial apoptosis, in response to treatment with BH3 peptides that mimic the activity of pro-apoptotic proteins from the BCL-2 family (Fraser et al, 2018). Cells that release cytochrome c in response to moderate or weak pro-apoptotic stimuli are considered to be primed for apoptosis, whereas cells that require strong stimuli to trigger cytochrome c release are unprimed. Apoptotic priming has been previously shown to determine whether healthy and cancerous cells undergo apoptosis in response to cellular stress or damage (Chonghaile et al, 2011; Sarosiek et al, 2017; Spetz et al, 2022). BH3 profiling showed a substantial reduction in apoptotic priming of Opa1-KO cells relative to WT cells as indicated by decreased sensitivity to the BIM BH3 peptide, which can inhibit all pro-survival Bcl-2 family proteins and directly activate BAX and BAK (Kale et al, 2018; Fig. 4A). Equal doses of the BIM BH3 peptide induced less cytochrome c release from mitochondria in Opa1-KO cells than WT cells, indicating that stronger pro-death stimuli are required to induce apoptosis in cells lacking Opa1 (Fig. 4A). We also tested MEF sensitivity to common apoptosis-inducing agents by quantifying positivity for Annexin V, which binds phosphatidylserine on the surface of cells undergoing early apoptosis, after treatment. Based on the BH3 profiling results, we would expect less apoptosis after treatment of Opa1-KO cells than WT cells. We found that indeed this was the case: Opa1-KO MEF were more resistant to treatment with the topoisomerase inhibitor etoposide, DNA damaging agent doxorubicin, and pan-kinase inhibitor staurosporine at 24 h (Fig. 4B). Similar differences in sensitivity were also detected at 48 and 72 h, suggesting that retention of cytochrome c durably protected Opa1 KO cells from commitment to apoptosis (Appendix Fig. S4). Overexpression of Opa1 maintains a WT-level response to apoptotic stimuli. While s-Opa1* cells show levels of apoptotic priming between WT and Opa1-KO cells, the short-form is not sufficient to rescue normal apoptotic priming of MEF. In contrast, l-Opa1* cells behave similarly to the WT cells across most conditions, restoring apoptotic priming (Fig. 4A). These results suggest that these lines would not be as resistant to apoptosis-inducing agents as the Opa1-KO cells and is evident in the 24-h chemosensitivity data for etoposide, doxorubicin, and staurosporine treatment (Fig. 4B) as well as data at 48- and 72-h post-treatment (Appendix Fig. S4). The presence of a stable l-Opa1

population has been previously shown to be essential in apoptotic resistance (Merkwirth et al, 2008, 2012). Our results indicate Opa1 processing is not required for apoptotic priming. In Opa1-KO or s-Opa1* cells, wider CJ at steady-state conditions correlated with cells resistant to apoptosis.

To understand the impact of Opa1-dependent remodeling on mitochondrial calcium homeostasis and mitochondrial permeability transition pore (mPTP) opening, we performed mitochondrial calcium retention capacity (CRC) assay on MEF cell lines. Mitochondria from permeabilized l-Opa1* cells show similar CRC profiles to WT (Fig. 4C). In contrast, s-Opa1* cells and Opa1-KO cells are observed to be more resistant to mPTP opening (Fig. 4C,D) and require higher concentrations of $CaCl_2$ to induce the pore opening transition (Fig. 4B). The presence of vesicular cristae and wider CJs at steady-state conditions correlate with s-Opa1* and Opa1-KO cells resistant to calcium-induced mPTP opening. In contrast, Opa1-OE cells undergo mPTP transition at lower amounts of $CaCl_2$ stimulation. We note an elevated presence of mitochondrial calcium deposits in the Opa1-OE cell line (Fig. EV1H).

Next, we evaluated the mitochondrial respiratory fitness of all cell lines to assess how a key cristae-dependent function varies upon Opa1 perturbation. As previously observed, mitochondrial respiration is severely impaired in Opa1-KO cells (Zhang et al, 2011; Fig. 5A). Basal OCR levels for Opa1-OE cells are similar to WT (Fig. 5B). s-Opa1* and l-Opa1* cells both have basal oxygen consumption respiration (OCR) levels significantly lower than WT, as previously reported (Lai et al, 2020; Lee et al, 2020). Cell lines with lower basal OCRs, also display significantly lower maximal respiration and spare capacity levels (Fig. 5B). Longer cristae and more amorphous cristae were observed in these cell lines. These results suggest that balanced levels of both forms of Opa1 are required for normal mitochondrial respiration.

Mitochondrial fitness was also measured in the $oma1^{-/-}$ background in which the l-Opa1* cells were generated (Wang et al, 2021). Like l-Opa1* cells, $oma1^{-/-}$ cells also have WT levels of cytochrome c release upon treatment with BIM (Appendix Fig. S5A). Additionally, $oma1^{-/-}$ MEF cells show WT-like propensity for mPTP opening under $Ca^{2+}$ stimulation albeit displaying a lower overall calcium buffering profile than WT and l-Opa1* cells (Appendix Fig. S5B). OCR values in l-Opa1* cells are significantly lower than in $oma1^{-/-}$ cells, indicating the $OMA1$ deletion is not the sole contributor to the impaired respiration observed in l-Opa1* cells (Appendix Fig. S5C,D). These data suggest that functional differences in the l-Opa1* mitochondria are dependent on the form of Opa1 present.

Finally, we investigated the effect of Opa1 imbalance on mtDNA maintenance. Mitochondrial fusion is essential for maintaining mtDNA stability (Elachouri et al, 2011) and Opa1 depletion has been shown to lead to a decrease in mtDNA number and translation (Chen et al, 2010). Consistent with previous reports, we observed robust mtDNA decrease in Opa1-KO MEF quantified by instant Structured Illumination Microscopy (iSIM) imaging (Appendix Fig. S6A–C) and qPCR (Appendix Fig. S6D). Stimulated emission depletion (STED) microscopy studies in HeLa and human fibroblast cells have shown that nucleoids occupy mitochondrial matrix spaces between clusters of cristae (Stephan et al, 2019) and when fusion is inhibited, nucleoids cluster without changes in size and copy number (Ramos et al, 2019). We observed higher nucleoid

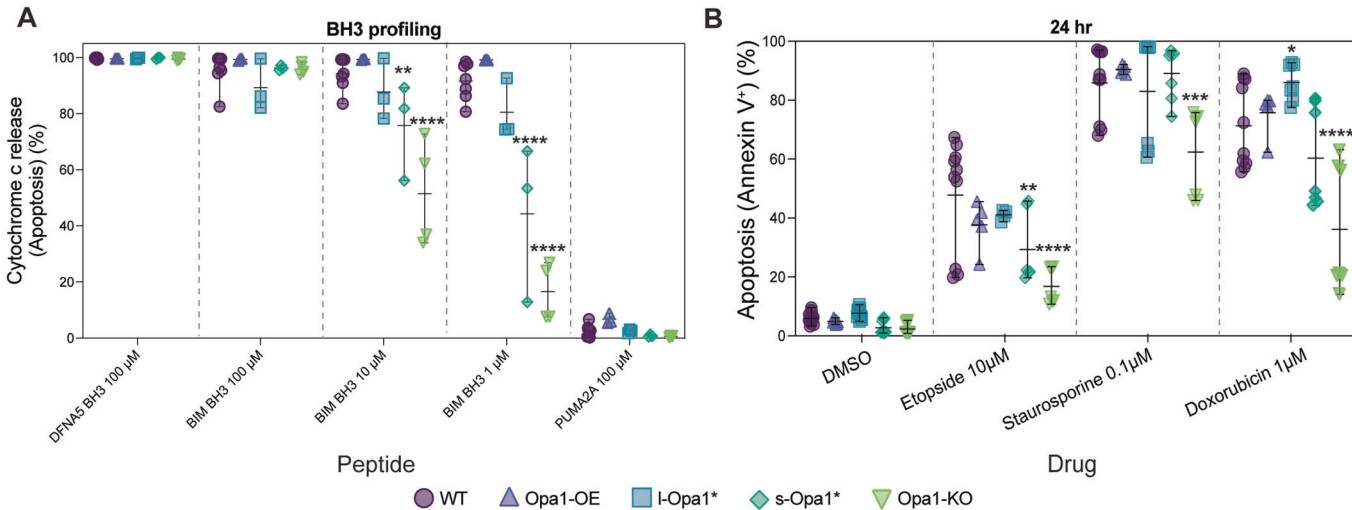

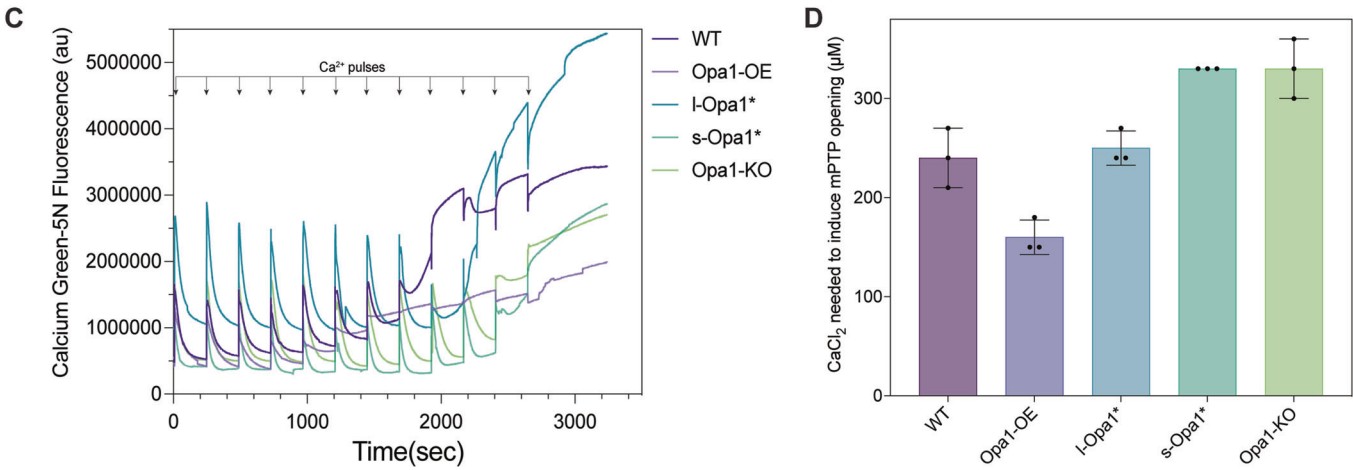

**Figure 4. l-Opa1* cells show WT apoptotic priming.**

(A) BH3 profiling of MEF lines with BIM BH3 peptide at indicated concentrations compared with positive control DFNA5 and negative control PUMA2A. Error bars represent range of $N = 3$–4 biological replicates (see methods for description). (B) MEF lines were treated with indicated agents for 24 h and apoptosis was detected via flow cytometry after staining with Annexin V. Error bars represent range of $N =$ minimum 4 biological replicates. (C) Representative traces of mitochondrial calcium retention capacity assays done in indicated MEF lines. (D) Quantification of CaCl2 concentration required to induce mPTP opening in (C). Error bars represent SD from $N = 3$ biological replicates. Data information: Significance of difference is tested relative to wild-type using the Holm–Sidak's multiple comparison test; *$p < 0.05$, **$p < 0.01$, ***$p < 0.005$, ****$p < 0.0001$. Source data are available online for this figure.

area and puncta in l-Opa1* cells than in WT cells (Appendix Fig. S6B,C). No difference in mtDNA copy number was observed in l-Opa1* cells by qPCR (Appendix Fig. S6D), suggesting these changes are likely due to mtDNA distribution and organization.

# Discussion

Understanding how sophisticated membrane architectures are generated in pleiomorphic organelles poses a challenge and opportunity for structural cell biology. The mitochondrion is an exemplar case where the organelle's morphology can take on a wide range of forms regulated by dynamic protein assemblies (Mageswaran et al, 2023). In

this work, we used state-of-the-art in situ cryo-ET to analyze how Opa1 processing regulates mitochondrial ultrastructure, in particular cristae morphology. Previous analyses of mitochondrial cristae morphology have been informed by conventional TEM micrographs of fixed and heavy metal-stained cells known to perturb membrane structures and introduce imaging artifacts (Scorrano et al, 2002; Olichon et al, 2003; Frezza et al, 2006). In overcoming these limitations, we characterized and quantified a diverse range of cristae morphologies specific to the form of Opa1 present in the cell. Here, we discuss structure/function relationships which emerged from this study (Fig. 6).

Our 3D data captures the diversity of cristae morphologies and can distinguish lamellar and tubular cristae. While the majority

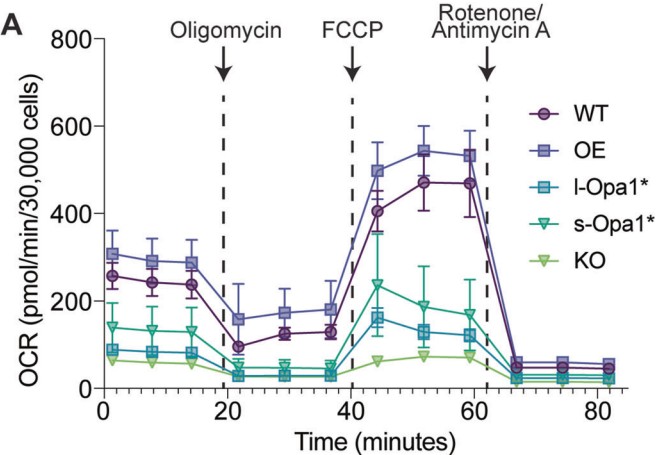
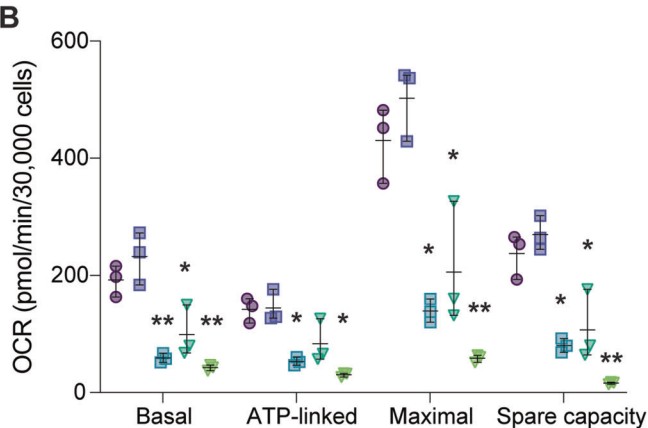

**Figure 5.  Defects to mitochondrial functions are observed in cells with altered Opa1 levels.**

(A) OCR plotted against time with the addition of each compound indicated by an arrow for oligomycin (2 µM), FCCP (1 µM), and rotenone/antimycin A (0.5 µM). Error bars represent SD from N = 3 biological replicates. (B) Aspects of mitochondrial respiration; basal respiration rates, the amount of respiration used for ATP production, maximum respiration and spare capacity, are extracted by the data plotted in (A). Error bars represent range for N = 3 biological replicates. Data Information: Significance of difference is tested relative to WT using Welch's t test; *p < 0.05, **p < 0.01. Source data are available online for this figure.

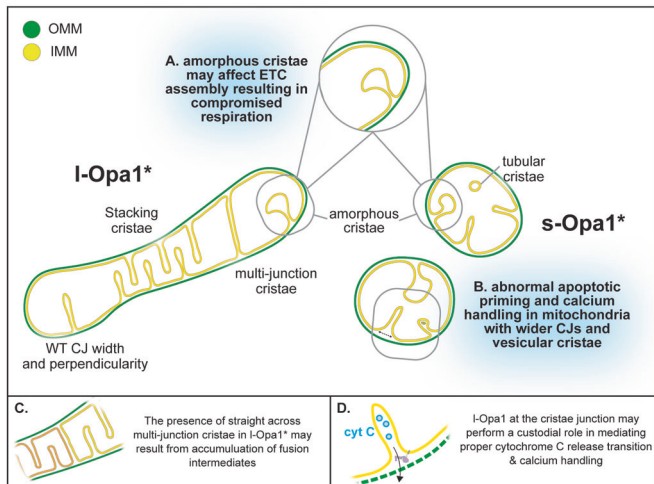

**Figure 6.  Summary of cryo-ET cristae observations.**

Cartoon schematic summarizing mitochondrial morphological observations dependent on Opa1 state. Mitochondria in l-Opa1* cells displayed WT-like CJ widths and perpendicularity. The majority of mitochondria displayed a stacking phenotype and multijunction cristae. Tubular and vesicular cristae were found in s-Opa1* cells. Summaries of functional dysfunctions corresponding to observed cristae morphological changes. (A) Respiratory defects correlate with a large number of unstructured cristae. (B) Defects in cytochrome c (CytC) release properties (as evaluated by BH3 profiling) and calcium handling correlate with wider CJ and vesicular cristae. (C) Cartoon scheme with hypothesized modes of action for l-Opa1 in the crista stacking. (D) Cartoon schematic with hypothesized modes of action for l-Opa1 in BH3 and calcium handling process.

of cristae are lamellar, tubular cristae were observed in WT mitochondria and a fraction of s-Opa1* cells, but are entirely absent from l-Opa1* cells (Figs. 2A,B,E and 6A,B). Tubular cristae are compatible with reported in vitro helical assemblies of the yeast homolog, s-Mgm1 reconstituted on membranes (Faelber et al,

2019). Recent single particle cryo-EM helical reconstructions of human s-Opa1 decorating the exterior of membranes tubes show how paddle domain interactions facilitate self-assembly (Malsburg et al, 2023; Nyenhuis et al, 2023). The increase in globular cristae in the Opa1-KO condition may imply that the presence of either or both forms of Opa1 support lamellar cristae. Indeed, a recent study found knock-down of Opa1 altered the ratio of lamellar to tubular cristae (Suga et al, 2023). Stacking cristae phenotype in l-Opa1* as well as increased presence of multijunction cristae may implicate an accumulation of fusion intermediates (Fig. 6C). Further integrative structural studies will also be necessary to directly relate how specific protein conformational states influence such morphologies (Fig. 6C; Harner et al, 2016).

Visualizing mitochondrial membrane structure using cryo-ET enables us to make accurate measurements of membrane distances. l-Opa1* cells maintained WT-like CJ widths (Fig. 3A), which with our BH3 profiling assay suggest that l-Opa1 plays a role in maintaining CJ widths and mediating the response to apoptotic stimulation (Fig. 6D). TEM imaging showed that cytochrome c release (stimulated by pro-apoptotic signaling proteins such as BID) results in the widening of cristae junctions and opening of the cristae lumen (Scorrano et al, 2002; Frezza et al, 2006). In contrast to previous studies, we investigated cristae state prior to any exposure to extracellular stress or apoptotic stimuli (Merkwirth et al, 2008, 2012). Building upon previous studies, we observed wider cristae junctions in Opa1-KO and s-Opa1* cells prior to apoptotic initiation (Fig. 3). Together, with our apoptotic priming results demonstrating that cells lacking l-Opa1 released less cytochrome c upon apoptosis stimulation, our observations support a specific CJ regulatory role for l-Opa1 in the cytochrome c release transition (Fig. 6B,C; Merkwirth et al, 2008).

Our findings that Opa1-KO cells are resistant to apoptosis may seem counterintuitive given previous reports of apoptotic resistance in Opa1-overexpressing cells (Frezza et al, 2006). However, this apparent discrepancy can be reconciled if considering multiple

roles for Opa1 during apoptotic membrane transitions. During apoptosis, Opa1 maintains cristae junctions, which may restrict cytochrome c release in response to pro-apoptotic signals (Frezza et al, 2006). However, the importance of rearrangement and disassembly of Opa1 complexes during apoptosis initiation, to facilitate CJ opening, as discussed above, demonstrates that Opa1 also has a direct role in the earlier steps when apoptosis is being initiated (Fig. 6D). This view is consistent with our finding that complete knock-out of Opa1 impairs cytochrome c release and can be protective against apoptosis-inducing agents (Fig. 4A,B). Our data indicate that apoptotic response is independent of Opa1 processing (Cipolat et al, 2006) and cytochrome c release is dependent on initial CJ widths.

Our BH3 profiling results are supported by CRC assay results, where l-Opa1 presence in MEF allows for WT-like responses towards increased $Ca^{2+}$ and apoptotic stimulation (Fig. 4C,D). These independent functional outcomes are connected by the importance of membrane integrity in mPTP opening (Strubbe-Rivera et al, 2021; Bernardi et al, 2023) and BAX/BAK-mediated cytochrome c cristae remodeling (Scorrano et al, 2002; Renault et al, 2015). Consistent with previous reports of calcium phosphate deposits, we observed dense deposits in the mitochondria matrix of most Opa1-OE mitochondria (Wolf et al, 2017), which correlates with a lower mPTP transition $CaCl_2$ threshold. Cryo-ET enabled this finding and allows us to attribute differences in density in the mitochondrial matrix to biological material, and not artifacts from chemical agents. We note a less condensed matrix in mitochondria in s-Opa1* cells and a lighter matrix in the Opa1-KO mitochondria, which we speculate may be related to matrix components other than calcium.

Our measurements also suggest that l-Opa1 may contribute to maintaining perpendicular CJ, as fewer perpendicular CJs were observed in s-Opa1* cells. Perpendicular CJs can facilitate cristae stacking, which we noticed in a striking number of l-Opa1* cells. Stacking cristae have been previously observed both under conditions that induce metabolic and endoplasmic reticulum stress upon addition of thapsigargin (Barad et al, 2022) and in $oma1^{-/-}$ cells (Anand et al, 2014). We speculate that cristae stacking may have features in common to an intermediate preceding fusion (Fig. 6C). Consistent with this role, we also note a correlation between stacked cristae organization in the l-Opa1* cells facilitating mtDNA distribution. While Opa1-KO cells have a dramatic loss of mtDNA number, we still visualize the presence of cristae in these mitochondria. This further supports emerging evidence that the organization of multiple crista contribute to mtDNA maintenance (Stephan et al, 2019; Jakubke et al, 2021).

We also assessed the effects of Opa1 state on mitochondrial respiration, a central function dependent on cristae shape (Cogliati et al, 2016). In short, compromised respiration observed in l-Opa1*, s-Opa1*, and Opa1-KO cells correlates with a larger percentage of amorphous cristae and longer cristae lengths. This suggests both forms of Opa1 play a role in maintaining a defined cristae shape and length. Our work indicates more subtle cristae phenotypes (such as vesicular and amorphous) may also not be optimal for respiration (Fig. 6). We speculate disruption of cristae shape could destabilize respiratory complex assembly in the cristae membranes or vice versa. Future studies will be necessary to more directly link in situ mitochondrial ultrastructure with mitochondrial dysfunction.

In summary, here we characterize cristae morphologies related to the levels or forms of Opa1 present in the cell. We characterize and quantify cristae morphological differences and define distinct differences specific to each Opa1 form. Notably, we find evidence that l-Opa1 plays important roles in maintaining CJ width and connectivity, which correlates well with WT-like apoptotic and calcium responses. We also demonstrate that both forms of Opa1 are required for maintaining cristae membrane shape and mitochondrial respiration, linking amorphous cristae with mitochondrial dysfunction. This work, in describing cristae morphology and functions, opens new opportunities for understanding mitochondrial ultrastructure and function, and motivates further studies visualizing and dissecting mechanisms underlying cristae heterogeneity by new live cell imaging approaches to gain insights into the spatio-temporal regulation of their lifecycle.

# Methods

## Cell lines and culture

MEF were maintained in DMEM supplemented with 10% fetal bovine serum (FBS) and 1% penicillin/streptomycin at 37 °C and 5% $CO_2$. Wildtype and Opa1 knock-out MEF cell lines were purchased from ATCC. $\Delta exon5b/oma1^{-/-}$ cells and Opa1 knock-outs transfected retrovirally with Opa1 Isoform 5 (KO Opa1 + Iso5) were kind gifts from David Chan (California Institute of Technology) (Song et al, 2007). The KO Opa1 + Iso5 were maintained in the same condition described above and supplemented with puromycin at 1 µg/mL. Cell genotypes were confirmed by Western blotting. To mediate a constitutive over-expression of Opa1 from the safe harbor locus Rosa26 in MEF, mouse Opa1 sequence was cloned into pR26-CMVconst (Addgene plasmid #127373; http://n2t.net/addgene:127373; RRID: Addgene_127373; kind gift by Lance Miller) to generate pR26-CMV-Opa1 (pAH33). Following sequence verification, the construct was co-transfected with pX330-sgR26 (Addgene plasmid #127376; http://n2t.net/addgene:127376; RRID: Addgene_127376; kind gift by Lance Miller) into WT MEF cells using Lipofectamine 3000. Transfected cells were selected with 5 µg/mL puromycin starting 24 h post-transfection for 7 days. Stable clones were expanded for 7 days, and their genotypes were confirmed by Western blotting. Opa1-OE MEF cell line was cultured in the same manner described above, with the addition of 2.5 µg/mL puromycin in the culture media.

Human HEK 293T cells (American Type Culture Collection; ATCC) were cultured in Dulbecco's Modified Eagle Medium (DMEM) supplemented with 10% heat-inactivated FBS (HI-FBS) and 1% penicillin–streptomycin. Samples of supernatant media from cell culture experiments were analyzed monthly for the presence of mycoplasma using MycoAlert PLUS (Lonza).

To generate a HEK 293T cell line bearing an OPA1 exon 5b knock-out, SpCas9 sgRNAs were cloned into pUC19-U6-BsmBI_cassette-SpCas9gRNA (BPK1520; Addgene ID 65777) harboring spacer sequences GCTCATTGTGAACTCGTGGCA (plasmid CJT90), GCCAACAGAAGCGCAAGGTGA (plasmid CJT91), GTTCTCCTCATTGTGAACTCG (plasmid CJT92), and GCAGAAGCGCAAGGTGATGGA (plasmid CJT93), each with added 5'Gs. Transfections were performed 20 h following seeding

of $2 \times 10^4$ HEK 293T cells per well in 96-well plates. Transfections contained 70 ng of SpCas9 nuclease plasmid (pCMV-T7-SpCas9-P2A-EGFP (RTW3027; Addgene ID 139987) and 15 ng each of two sgRNA expression plasmids mixed with 0.3 μL of TransIT-X2 (Mirus) in a total volume of 15 μL Opti-MEM (Thermo Fisher Scientific), incubated for 15 minutes at room temperature, and distributed across the seeded HEK 293T cells. Cells were grown for approximately 72 h prior to extracting genomic DNA (gDNA) by discarding the media, resuspending the cells in 100 μL of quick lysis buffer (20 mM Hepes pH 7.5, 100 mM KCl, 5 mM MgCl2, 5% glycerol, 25 mM DTT, 0.1% Triton X-100, and 60 ng/μL Proteinase K (New England Biolabs; NEB)), heating the lysate for 6 min at 65 °C, heating at 98 °C for 2 min, and then storing at −20 °C. Editing efficiency in bulk transfected cells was assessed by next-generation sequencing (NGS) essentially as previously described (Walton et al, 2020) using PCR round 1 primers oLLH9-ACACTCTTTCCCTACACGACGCTCTTCCGATCTCATCTGTT CCTTTGTTGCACCCTTGG and oLLH10-GACTGGAGTTCAGA CGTGTGCTCTTCCGATCTGGAGTCCATGAACAGATTGAGG TGAC. To create cell lines, HEK 293T cells were seeded and transfected with plasmids RTW3027 and both sgRNAs CJT91 and CJT92. Transfected cells were grown for approximately 72 h prior to dilution plating into 96-well plates and grown until confluent. Cells were transferred into 24-well plates with some cell mass reserved to extract genomic DNA (gDNA) for genotyping via PCR and NGS to verify biallelic *OPA1* exon 5b deletion between the two SpCas9-sgRNA cleavage sites.

To generate a HEK 293T cell line bearing an *OPA1* R194G mutation, SpCas9 sgRNAs were cloned into BPK1520 harboring spacer sequences GCGGCGTTTAGAGCAACAGAT (plasmid CJT87), GCGTTTAGAGCAACAGATCGT (plasmid CJT88), and GCGTTTAGAGCAACAGATCG (plasmid CJT89). Adenine base editor (ABE) plasmids included pCMV-T7-ABE8e-nSpCas9-P2A-EGFP (KAC978; Addgene ID 185910) or pCMV-T7-ABE8e-nSpG-P2A-EGFP (KAC984; Addgene ID 185911). Transfections were performed 20 h following seeding of $2 \times 10^4$ HEK 293T cells per well in 96-well plates and contained 70 ng of ABE8e plasmid and 30 ng of sgRNA expression plasmid mixed with 0.3 μL of TransIT-X2 (Mirus) in a total volume of 15 μL Opti-MEM (Thermo Fisher Scientific). The transfection mixtures were incubated for 15 min at room temperature and distributed across the seeded HEK 293T cells. Cells were grown for approximately 72 h prior to extracting gDNA as described above. Editing efficiency in bulk transfected cells was assessed by NGS using PCR round 1 primers oLLH7- ACACTCTTTCCCTACACGACGCTCTTCCGATCTGCT TAGGCTGTTGACATCACTGGAGAATG and oLLH8-GACTG GAGTTCAGACGTGTGCTCTTCCGATCTCCAGAACTGCCAC GTAATACCTTGTAC. To create cell lines, HEK 293T cells were seeded and transfected with plasmids KAC984 and CJT88. Transfected cells were grown for approximately 72 h prior to dilution plating into 96-well plates and grown until confluent. Cells were transferred into 24-well plates with some cell mass reserved to extract genomic DNA (gDNA) for genotyping via PCR and NGS to verify biallelic introduction of *OPA1*-R194G.

## Live cell epifluorescence microscopy

Confluent MEF cells were harvested, seeded onto 35 mm glass-bottom dishes (MatTek Life Sciences) coated with poly-D-lysine

(0.1 mg/mL) and allowed to grow overnight at 37 °C under 5% $CO_2$. For visualization of mitochondria, cells were stained with 50 nM MitoTracker™ Deep Red FM (Thermo Fisher Scientific) at 37 °C for 15 min. Following three rounds of washes with 1X PBS, cells were placed in Live Cell Imaging Solution (Invitrogen). Imaging relating to Fig. EV3 was performed using Zeiss Axio Observer Z1 Advanced Marianas™ Microscope system, an Alpha Plan-Apochromat ×100/1.46 NA Oil TIRF Objective M27 and Prime 95B scientific CMOS camera (Photometrics). MitoTracker™-stained mitochondria were imaged using "Cy5" filter set (Cy5-4040C, Excitation: 628/40 nm [608-648 nm], Emission: 692/40 nm [672–712 nm], Dichroic Mirror: 660 nm) (Semrock). Temperature, humidity, and $CO_2$ concentrations were controlled with an Okolab Microscope Stage Incubator System. Image acquisition and processing were done using SlideBook™6 (Intelligent Imaging Innovations, Inc., Denver, CO) and Fiji (Schindelin et al, 2012). Time-lapse videos of stained mitochondria were taken at one frame per 30 s for a duration of 5 min.

## Fluorescence imaging and quantification

To determine mitochondrial network morphology, MEF cells were seeded onto glass-bottom dishes and grown to ~80% confluency. Cells were washed with PBS and fixed in 4% paraformaldehyde (PFA) in PBS for 20 mins. Following three washes with PBS, cells were permeabilized in 0.1% Triton X-100 diluted in PBS for 5 min. Cells were incubated in 2% normal goat serum and 0.05% Triton X-100 in PBS for 1 h, and subsequently incubated in the same buffer with 1:300 Tom20 Recombinant rabbit antibody (Invitrogen) overnight at 4 °C. After thorough washing with PBS, cells were incubated with 1:1000 goat anti-rabbit Alexa Fluor® 488 AffiniPure antibody (Jackson ImmunoResearch) for 1 h at RT. Cells were washed in PBS and coated with UltraCruz® Aqueous Mounting Medium with DAPI (Santa Cruz) prior to imaging. Images were captured using A1R HD25 point scanning confocal with GaAsP and PMT detectors, equipped with an Apo TIRF 60x/1.49 NA objective lens and Ti2 Z-drive. Cells were manually scored into four morphological classifications (Wang et al, 2021): "Fragmented" refers to cells that contain spherical mitochondrial fragments with two or less short tubules present. "Short tubular" refers to cells containing a mixture of fragmented and short tubular mitochondria. "Long tubular" refers to cells with elongated mitochondria, but not fused into a mitochondrial mesh. "Interconnected" refers to cells with a highly interconnected network of mitochondrial filaments, with few isolated mitochondria present.

## Cryo-EM specimen preparation

Cells were prepared following the deposition method. Cells were detached and counted with a hemocytometer. Quantifoil 200 mesh holey carbon R2/2 (EMS) were glow-discharged for 60 s or 90 s at 20 mA or 15 mA using a PELCO easiGlow glow discharge system (Ted Pella). Approximately 1000–3000 cells were deposited onto a grid by pipetting 3 μL of detached cells onto the EM grid. Blotting and plunging was performed in a FEI Vitrobot Mark IV (Thermo Fisher Scientific (TFS)) at RT, 100% humidity with a waiting time of 60 s, one-side blotting time of 15 s and blotting force of 10 or 7. Customized parafilm sheets were used for one-sided blotting. All

subsequent grid handling and transfers were performed in liquid nitrogen. Grids were clipped onto cryo-FIB autogrids (TFS).

## Cryo-FIB milling

Grids were loaded in an Aquilos 2 Cryo-FIB (TFS). Specimen was sputter coated inside the cryo-FIB chamber with inorganic platinum, an integrated gas injection system (GIS) was used to deposit an organometallic platinum layer to protect the specimen surface and avoid uneven thinning of cells (Wagner et al, 2020). Cryo-FIB milling was performed on the specimen using two rectangular patterns to mill top and bottom parts of cells, and two extra rectangular patterns were used to create macro-expansion joints to improve lamellae instability (Wolff et al, 2019). Cryo-FIB was performed at a nominal tilt angle of 14-26, which translates into a milling angle of 7-19. Cryo-FIB milling was performed in several steps of decreasing ion beam current ranging from 1 nA to 10 pA and decreasing thickness to obtain 200–400 nm lamellae.

## Cryo-electron tomography

All imaging was performed on a FEI Titan Krios (TFS) transmission electron microscope operated at 300KeV equipped with a Gatan BioQuantum K3 energy filter (20 eV zero-loss filtering and a Gatan K3 direct electron detector. Prior acquisition, a full K3 gain reference was acquired, and ZLP and BioQuantum energy filter were finely tuned. The nominal magnification for data collection was ×33,000 giving a calibrated 4 K pixel size of 2.758. Data collection was performed in nanoprobe mode using SerialEM (Mastronarde, 2003) or TFS Tomography 6 software. The tilt range varied depending on the lamella, but generally was from −70 to 70 in 2 steps following the dose-symmetric scheme (Hagen et al, 2017). Tilt images were acquired as $8 K \times 11 K$ super-resolution movies of 6 frames with a set dose rate of $1.5–3 e^−/Å/s$. Tilt series were collected at a range of nominal defoci between −3.5 and −5 μm and a target total dose of $100–180 e^−/Å$ (Table EV1).

## Cryo-electron tomography image processing

Acquired tilted super-resolution movies were motion corrected and Fourier cropped to $4 K \times 5 K$ stacks, minimizing aliasing effects using *framealign* from IMOD (Kremer et al, 1996). Tilt-series were aligned using etomo in IMOD (Mastronarde and Held, 2017). CTF-estimation was performed in IMOD (Turoňová et al, 2017) and/or using customized MATLAB scripts. CTF-correction was performed by *ctfphaseflip* program in IMOD. CTF-corrected unbinned tomograms were reconstructed by weighted back projection with and without a SIRT-like filter and subsequently 2×, 4× and 8× in IMOD. Cryo-electron tomograms were denoised using Topaz (Bepler et al, 2020) and summed projection of cryo-tomogram slices were performed in *Dynamo* (Castaño-Díez et al, 2012) complemented with customized MATLAB scripts.

## 3D segmentation

Segmentation was done in TomoSegMemTV (Martinez-Sanchez et al, 2014) to create the first triangulation of mitochondrial membranes. Such triangulation was refined using Amira (TFS) by

unbiased semi-automatic approaches. Final triangulated surfaces were remeshed and smooth in Amira for final rendering.

## Quantitative analysis of cryo-ET data

### Mitochondrial shape
Mitochondria morphology was categorized into "ellipsoidal," "round," "heart-shaped" (when displaying a polygon shape) and "partial" (when mitochondria was out of the XY image) by visual inspection of cryo-electron tomograms.

### Mitochondrial size
Mitochondria were outlined in summed projection images of the central slices of cryo-electron tomograms in FIJI using the "polygon selection" tool and pressing the measure key to output the area of outline mitochondria in $nm^2$.

### Mitochondrial coverage
Mitochondrial area in $μm^2$ obtained from mitochondria size measurements was divided by the total area of the summed projected image.

### Matrix density
Mitochondria density was measured in summed projection images of the central slices of cryo-electron tomograms that were all equally gray scale normalized in FIJI by applying the function equalize histogram set at 0.35% for all images. Three lines were drawn in the matrix region of the mitochondria under analysis and their mean gray value was calculated by pressing the measure button in FIJI (Schindelin et al, 2012). Three measurements per mitochondria were obtained, thus, the mean was calculated to obtain a single value per mitochondrial matrix.

### Cristae density
Number of cristae was quantified in cryo-electron tomogram using the multi-point tool in FIJI. The number of cristae was normalized against area of mitochondria in $μm^2$.

### Mitochondrial volume
Total mitochondria volume was calculated in Amira by summing the volume of cristae lumen (CL), inter membrane spacing (IMS) and matrix volume in $μm^3$. CL, IMS and matrix volumes was outputted by Amira based on the 3D surface of each compartment segmented and rendered in Amira with the module "measure surface." Ratios were calculated by dividing the volume values of the specified mitochondrial compartments.

### Cristae directionality
Cristae was classified as "straight," "tilted," or "disconnected" by visual inspection of cryo-electron tomograms.

### Cristae shape
Cristae was classified as "lamellar," "globular," "tubular," or "unusual" by visual inspection of cryo-electron tomograms. Within the category "unusual" the following classes were defined: "loop" when cristae present two cristae connection with the IMS and was curved, "pinching" when cristae membranes presenting punctual touching points, "straight-across" when cristae present two cristae connection with the IMS just opposite to each other forming a

straight septum-like structure across a mitochondrion, "amorphous" when cristae displayed an irregular polygon shape, "splitting" when cristae branched into two or more cristae within a giving mitochondrion, "ring" when cristae formed a circular ring within mitochondria, "zip" when cristae membranes come close and only one membrane was distinguished that later opens up into regular lamellar cristae, and "vesicular" when cristae was wide, usually amorphous, but contained electron dense material resembling to membranes.

### Cristae length

Cristae length was measured in cryo-electron tomograms by extracting the cristae volumes in *Dynamo* using the "oblique slices in tomoslice" tool (https://wiki.dynamo.biozentrum.unibas.ch/w/index.php/Oblique_slices_in_tomoslice). Then, length was computing using the length tool in *Dynamo* (https://wiki.dynamo.biozentrum.unibas.ch/w/index.php/Walkthrough_on_GUI_based_tilt_series_alignment_(EMBO2021)#Visualization_matrix).

### Cristae width

Subtomogram averaging was performed in *Dynamo*. Particles were manually identified using "*dtmslice*" interface in *Dynamo* (Navarro et al, 2018, 2020). Subtomograms with a size of $(1058.8)^3$ Å were extracted from 4×-binned tomograms. An initial reference generated from a random set of particles was used for 3D particle alignment. A total of 12 iterations were used to align particles until convergence, i.e., until no further improvement of alignment parameters was detected by additional iterations, and then final averages were obtained. Final averages were generated from 222 (WT), 430 (OE-Opa1), 323 l-Opa1*, 653 s-Opa1* and 243 KO-Opa1 subtomograms. EM densities were visualized in UCSF Chimera (Pettersen et al, 2004). Cristae width is measured per particle in 3D width measurement was done in *Dynamo* by cross-correlation of each particle against a set of 40 templates displaying a distance range between membranes of 1 to 40 pixels (corresponding to 2.2–88 nm distance, https://wiki.dynamo.biozentrum.unibas.ch/w/index.php/Framework_for_estimation_of_membrane_thickness and https://github.com/NavarroPP/membraneThickness/blob/main/CristaeThickness.m). A cross-correlation peak per particle is outputted corresponding to the distance value between the two cristae membranes defined here as cristae width (Appendix Fig. S3).

### Cristae junction measurement

Cristae junctions were measured in summed projection images of 10 slices from each tomogram. Each CJ was isolated, and the width was measured using the line tool and measurement function in FIJI (Schindelin et al, 2012). The angle of each CJ was measured using the angle tool and measure function in FIJI. If a CJ was visible in multiple and nonoverlapping sections of the tomograms, multiple measurements were made for that CJ and averaged to represent the overall 3D shape of the CJ.

### Statistical analysis

Statistical tests performed to assess differences in our quantifications were chosen based on the distribution of the data points. Since most of our data did not follow a normal distribution, we used non-parametric tests as Mann–Whitney test in order to measure significant difference between two data sets (e.g., WT vs Opa1-OE). When both data sets followed normal distribution

unpaired *t* test with Welch's correction was used since standard deviations were different between the compared data sets. Furthermore, additional statistical tests were applied to reassure statistical results outputting very similar p values and same significance. All statistical analyses were performed in Prism 9 GraphPad software.

### Mitochondrial respiration assays

Cells were seeded in a Seahorse XFe96 Cell Culture plate in 80 μL of DMEM supplemented with 10% FBS and 1% penicillin/streptomycin at a concentration of 30,000 cells/well and left to recover at 37 °C in 5% $CO_2$ for 6 h. Afterwards cells were then washed three times with prewarmed assay buffer (XF Seahorse DMEM media supplemented with 10 mM Seahorse XF Glucose, 1 mM Seahorse XF Pyruvate, and 2 mM Seahorse XF L-Glutamine), covered with 180 μL of assay buffer, and incubated for one hour at 37 °C under no atmospheric $CO_2$. Prior to the assay, the injection ports on the sensor cartridge were loaded with 2.0 μM oligomycin, 1 μM carbonyl cyanide-*p*-trifluoromethoxyphenylhydrazone (FCCP), and 0.5 μM of rotenone/antimycin A. Before the injection of drugs, the Seahorse XFe96 Analyzer mixed the assay media for each well for 10 min and took three baseline measurements. After the injection of each drug, the analyzer mixed for 3 min, waited for 1 min, and the measured for 3 min, three times. Oxygen consumption rates (OCRs) are the average oxygen consumption rate during each three-minute measurement time and were normalized to cells/well. Biological replicates indicate assay performed on different flasks of MEF cell culture grown under the same condition and passage number. Statistical analysis performed to assess differences in our measurements were chosen based on the distribution of the data points. The mitochondrial respiration data followed a normal distribution by the Shapiro–Wilk normalcy test. We used unpaired *t* test with Welch's correction to evaluate significance as standard deviations were different between the compared data sets. All statistical analyses were performed in Prism 9 GraphPad software.

### BH3 profiling and chemosensitivity methods

BH3 profiling was conducted by flow cytometry according to published protocols (Fraser et al, 2018). Briefly, cells in culture were trypsinized and added to wells of prepared 96 well plates containing the indicated peptide conditions and 0.001% digitonin in mannitol experimental buffer (MEB; 10 mM HEPES (pH 7.5), 150 mM mannitol, 50 mM KCl, 0.02 mM EGTA, 0.02 mM EDTA, 0.1% BSA, and 5 mM succinate). Peptide treatments were carried out for 60 min at 28 °C, then cells were fixed for 10 min in 2% PFA. Fixation was quenched with N2 buffer (1.7 M tris base and 1.25 M glycine (pH 9.1)), then cells were stained overnight with DAPI and an Alexa Fluor 647-conjugated anti-cytochrome c antibody (Biolegend, clone 6H2.B4). Stained cells were analyzed using an Attune NxT flow cytometer, with gates drawn based on cytochrome c staining in the negative and positive control treatments (PUMA2A and DFNA5 peptide). The percentage of cytochrome c negative cells was reported for each peptide treatment condition.

   For chemosensitivity assays, cells were plated at $10^4$ cells per well in 100 μL culture medium on 96-well flat-bottom plates (Denville). They were treated with the following drugs at specified concentrations: etoposide 10 μM, staurosporine 0.1 μM, and doxorubicin

1 μM. After 24 h in standard tissue culture conditions, cells were stained with Alexa Fluor 488-conjugated Annexin V in 10× Annexin binding buffer [0.1 M Hepes (pH 7.4), 1.4 M NaCl, and 25 mM $CaCl_2$ solution]. Alexa Fluor 488-conjugated Annexin V was added to the solution at a 1:500 dilution. The staining solution was added to the cells at a 1:10 dilution, and the cells were allowed to stain for 20 minutes on ice in the dark. Annexin V positivity was measured by Attune flow cytometer equipped with an autosampler (Thermo Fisher Scientific). Biological replicates indicate assay performed on different flasks of MEF cell culture grown under the same condition and passage number. Statistical analysis was performed using a two-way ANOVA with Holm–Sidak's correction for multiple hypothesis since the data were normally distributed (Fraser et al, 2022; Singh et al, 2023). All statistical analyses were performed in Prism 9 GraphPad software.

### Mitochondrial calcium retention capacity (CRC) assays

MEF cells ($1 \times 10^6$) were incubated in 150 μl assay buffer (125 mM KCl, 20 mM HEPES, 2 mM $K_2HPO_4$, 5 mM glutamate, 5 mM malate, 5 mM succinate, 1 mM $MgCl_2$, 5 μM EGTA, 1 μM Calcium Green-5N, 40 μM digitonin, pH 7.2) supplemented with 1.5 μM thapsigargin (Sigma) for 10 min. Fluorescence was continuously monitored using a PerkinElmer EnVision plate reader (excitation, 485 nm; emission, 535 nm). Sequential 30 μM $CaCl_2$ pulses were administered until $Ca^{2+}$ uptake ceased, and a sudden release of previously taken up $Ca^{2+}$ indicated by a sustained increase in fluorescence reading, consistent with mPTP opening. Biological replicates indicate assay performed on different flasks of MEF cell culture grown under the same condition and passage number.

### Mitochondrial network and mtDNA imaging and analysis by instant structured illumination microscopy (iSIM)

For live-cell imaging, cells were stained with PKMito Orange probe (Spirochrome) and SYBR™ Gold Nucleic Acid Stain (Thermo Fisher Scientific), following manufacturer's instructions. 250 nM PKMito Orange and 1:10 000 SYBR™ Gold Nucleic Acid Stain were the optimal concentrations for the cell type. Cells were allowed to rest for 1–2 h before imaging. Instant structured illumination microscopy (iSIM) was performed on the custom-built microscope set-up at the EPFL Laboratory of Experimental Biophysics (York et al, 2013; Mahecic et al, 2020). Time-lapse images were acquired from MEF cells acclimatized in a 37 °C chamber, using 488-nm and 561-nm excitation lasers, a 1.49 NA oil immersion objective (APONXOTIRF; Olympus), and an sCMOS camera (PrimeBSI, 01-PRIME-BSI-R-M-16-C; Photometric), with additional fast vertically scanning piezo actuated mirrors for scanning/illumination pattern homogenization. The raw images were deconvolved using the Richardson Lucy algorithm as implemented by the flowdec Python package (Czech et al, 2019) with 10 iterations. Deconvolved iSIM images of stained MEF cells were processed and analyzed using the CellProfiler™ software (Stirling et al, 2021). Briefly, mitochondria were segmented from the PKMito Orange signal using the Robust Background thresholding algorithm with 2 standard deviations following median filtering, which was subsequently used to mask and segment the nucleoid foci from the SYBR™ Gold image (Gaussian filtering and Robust Background algorithm with 2 standard deviations). The size and shape of each segmented object were then measured, and each cell's values are displayed normalized to the segmented mitochondrial area. These data are not normally distributed, thus we used Mann-Whitney test to measure significant difference between WT and the other cell lines. All statistical analyses were performed in Prism 9 GraphPad software.

### Quantification of mtDNA by qPCR

Total genomic DNA was extracted using Qiagen DNeasy kit following the manufacturer's protocol and 30 ng DNA was used for each qPCR reaction. Two mitochondrial probes were each normalized to a nuclear probe to calculate mtDNA copy number relative to nuclear genome copies. This relative mtDNA copy number for all genotypes were then normalized to that of WT MEF. Biological replicates indicate assay performed on MEF cell culture derived from different flasks, grown under the same condition and passage number. Sequences of qPCR primers:

MT-16S Fwd: CCGCAAGGGAAAGATGAAAGAC
MT-16S Rev: TCGTTTGGTTTCGGGGTTTC
MT-ND1 Fwd: CTAGCAGAAACAAACCGGGC
MT-ND1 Rev: CCGGCTGCGTATTCTACGTT
nuclear HK2 Fwd: GCCAGCCTCTCCTGATTTTAGTGT
nuclear HK2 Rev: GGGAACACAAAAGACCTCTTCTGG

These data followed a normal distribution by the Shapiro–Wilk normalcy test. We used unpaired *t* test with Welch's correction to evaluate significance as standard deviations were different between the compared data sets. All statistical analyses were performed in Prism 9 GraphPad software.

## Data availability

All data, strains, and plasmids will be available upon request. Representative tomograms of each cell line are deposited in the EMDB—WT: EMD-43063, EMD-43064, EMD-43065, EMD-43066; Opa1-OE: EMD-43067, EMD-43068, EMD-43069, EMD-43070; l-Opa1*: EMD-43049, EMD-43050, EMD-43051, EMD-43052; s-Opa1*: EMD-43053, EMD-43054, EMD-43055, EMD-43056, EMD-53057; Opa1-KO: EMD-43058, EMD-43059, EMD—43060, EMD-43061, EMD-43062. Corresponding raw data are deposited in EMPIAR: EMPIAR-11820 (WT), EMPIAR-11818 (Opa1-OE), EMPIAR-11819 (l-Opa1*), EMPIAR-11826 (s-Opa1*), and EMPIAR-11817 (Opa1-KO).

## Peer review information

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

## Acknowledgements

We are grateful to Christopher Borsa, Phat Vinh Dip, and Edward Brignole at the MIT.nano cryo-EM facility and Kang Song and Chen Xu at the University of Massachusetts cryo-EM facility for providing access to the cryo-EM

microscopes and for all their help, advice, and maintenance of cryo-EM equipment. Electron microscopy was performed in the Microscopy Core of the Center for Systems Biology/Program in Membrane Biology, which is partially supported by an Inflammatory Bowel Disease Grant DK043351 and a Boston Area Diabetes and Endocrinology Research Center (BADERC) Award DK057521. We thank Connor Tou for assistance in cloning guide RNAs. We thank Dr Patrick Ward and Dr Owen Skinner for helpful discussions, and the laboratories of Dr Vamsi Mootha and Dr Joshua Kaplan at Department of Molecular Biology, MGH for equipment access and support. MYF is a Fellow of The Jane Coffin Childs Fund for Medical Research. PPN was supported by the Swiss National Science Foundation (SNSF) Early Postdoc.Mobility P2BSP3_188112 and Postdoc.Mobility P400PB_199252 fellowships. ZI was supported by NIA award F32AG077861. This work was supported by a Mass General Hospital ECOR Howard M. Goodman Fellowship (to BPK); Charles H. Hood Foundation Child Health Research Awards to LHC and KAS; and Alex's Lemonade Stand Foundation for Childhood Cancers Research Award to KAS. This work was supported by funding from the National Institutes of Health (R01DK125263 and R37CA248565 to KAS; R35GM142553 to LHC).

## Author contributions

**Michelle Y Fry**: Conceptualization; Data curation; Software; Formal analysis; Funding acquisition; Validation; Investigation; Visualization; Methodology; Writing—original draft; Writing—review and editing. **Paula P Navarro**: Conceptualization; Data curation; Software; Formal analysis; Supervision; Funding acquisition; Validation; Investigation; Visualization; Methodology; Writing—original draft; Writing—review and editing. **Pusparanee Hakim**: Data curation; Formal analysis; Validation; Investigation; Visualization; Methodology; Writing—original draft; Writing—review and editing. **Virly Y Ananda**: Data curation; Software; Formal analysis; Validation; Visualization; Methodology; Writing—review and editing. **Xingping Qin**: Data curation; Formal analysis; Validation; Investigation; Visualization; Writing—review and editing. **Juan C Landoni**: Data curation; Formal analysis; Validation; Investigation; Writing—review and editing. **Sneha Rath**: Data curation; Formal analysis; Validation; Investigation; Writing—review and editing. **Zintis Inde**: Data curation; Validation; Investigation; Writing—review and editing. **Camila Makhlouta Lugo**: Data curation; Formal analysis; Validation; Visualization; Methodology; Writing—review and editing. **Bridget E Luce**: Formal analysis; Validation; Visualization; Methodology; Writing—review and editing. **Yifan Ge**: Conceptualization; Data curation; Investigation; Writing—review and editing. **Julie L McDonald**: Data curation; Investigation; Writing—review and editing. **Ilzat Ali**: Software; Formal analysis; Visualization. **Leillani L Ha**: Formal analysis; Investigation; Methodology. **Benjamin P Kleinstiver**: Supervision; Funding acquisition; Writing—review and editing. **David C Chan**: Resources; Writing—review and editing. **Kristopher A Sarosiek**: Supervision; Funding acquisition; Writing—review and editing. **Luke H Chao**: Conceptualization; Supervision; Funding acquisition; Visualization; Methodology; Writing—original draft; Project administration; Writing—review and editing.

## Disclosure and competing interests statement

BPK is an inventor on patents and/or patent applications filed by Mass General Brigham that describe genome engineering technologies. He is a consultant for EcoR1 capital and is an advisor to Acrigen Biosciences, Life Edit Therapeutics, and Prime Medicine. LHC is an advisor for Stealth Biotherapeutics. The remaining authors declare that there are no competing financial interests.

# Expanded View Figures

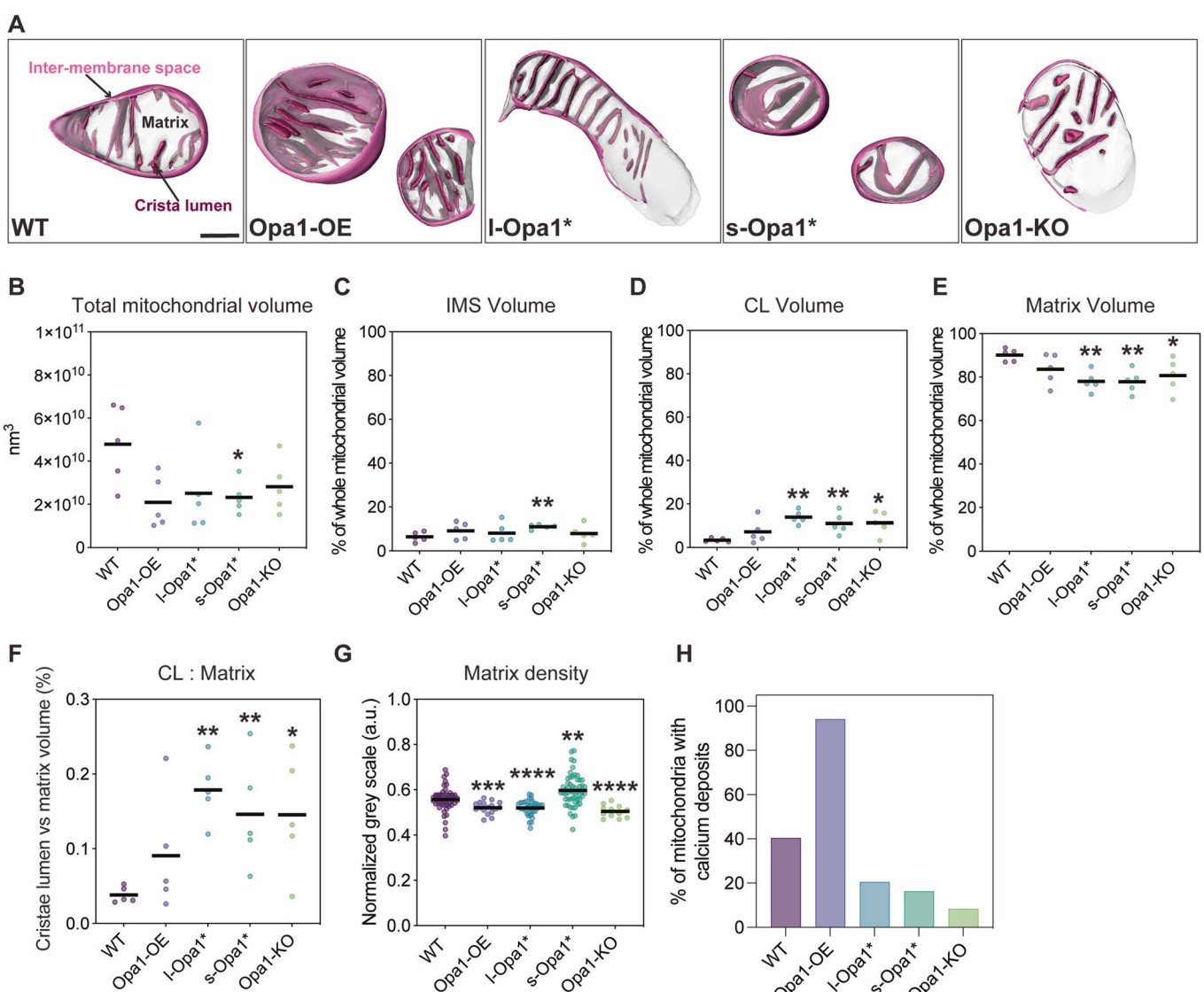

**Figure EV1. Mitochondrial subcompartment volumes.**

(A) Three-dimensional renderings of segmented inter-membrane space (IMS, pink surface), cristae lumen (CL, magenta surface), and matrix (translucent gray surface) volumes. Scale bar = 200 nm. (B) Total mitochondrial volume across indicated cell lines. $N = 5$ cells for all cell lines. (C) Quantification of IMS volume relative to total volume of each mitochondrion indicated in (B). $N = 5$ cells for all cell lines. (D) Quantification of CL volume relative to total volume of each mitochondrion indicated in (B). $N = 5$ cells for all cell lines. (E) Quantification of matrix volume relative to total volume of each mitochondrion indicated in (B). $N = 5$ cells for all cell lines. (F) CL to matrix ratio across cell lines. $N = 5$ cells for all cell lines. (G) Normalized gray scale mitochondrial matrix value across cell lines. $N = 5$ cells for all cell lines. (H) Graph bar representing percentage of cells with detected calcium deposits in cryo-electron tomograms. $N$ refers to the number of mitochondria: WT = 57, Opa1-OE = 17, l-Opa1* = 39, s-Opa1* = 55, Opa1-KO = 12. Data information: Scatter plots show data distribution, the mean is shown by a bold black line. Significance of difference is tested relative to wild type using Mann–Whitney test in (B, D, E, G); $*p < 0.05$, $**p < 0.01$, $***p < 0.001$, $****p < 0.0001$; and unpaired $t$ test in (C): $**p < 0.01$.

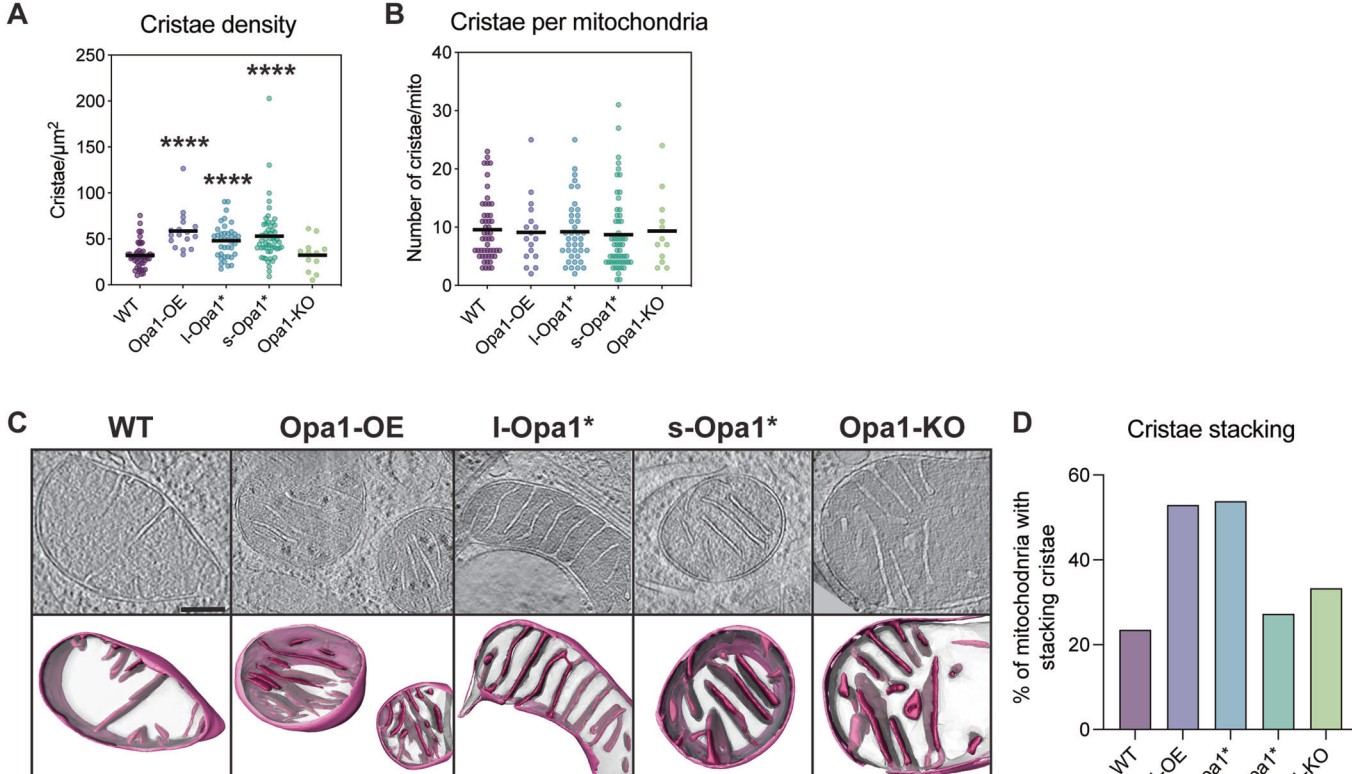

**Figure EV2. Cristae analysis.**

(**A**) Cristae density (cristae per µm²) represented as a scatter plot. *N* refers to number of cells; WT = 33, Opa1-OE = 7, l-Opa1* = 21, s-Opa1* = 28, Opa1-KO = 11. (**B**) Number of cristae per mitochondria represented as a scatter plot. N refers to number of cells; WT = 51, Opa1-OE = 17, l-Opa1* = 39, s-Opa1* = 55, Opa1-KO = 12. (**C**) (Top) Summed, projected central slices of cryo-electron tomograms visualizing mitochondria with stacking crista characteristics, supported by 3D representations consisting of their sub compartments (bottom) in indicated MEF lines. Scale bar = 200 nm. *N* refers to number of cells; WT = 57, Opa1-OE = 17, l-Opa1* = 39, s-Opa1* = 55, Opa1-KO = 12. The representative tomograms for WT, Opa1-OE, l-Opa1*, and Opa1-KO are the same as in Fig. 1A. The representative s-Opa1* tomogram is the same as the second from the left in Appendix Fig. S2A. (**D**) Graph bar representing percentage of mitochondria with stacking crista formation in each MEF line. *N* refers to number of cells; WT = 57, Opa1-OE = 17, l-Opa1* = 39, s-Opa1* = 55, Opa1-KO = 12. Data information: Significance of difference is tested relative to WT using Mann–Whitney test; ****$p < 0.0001$.

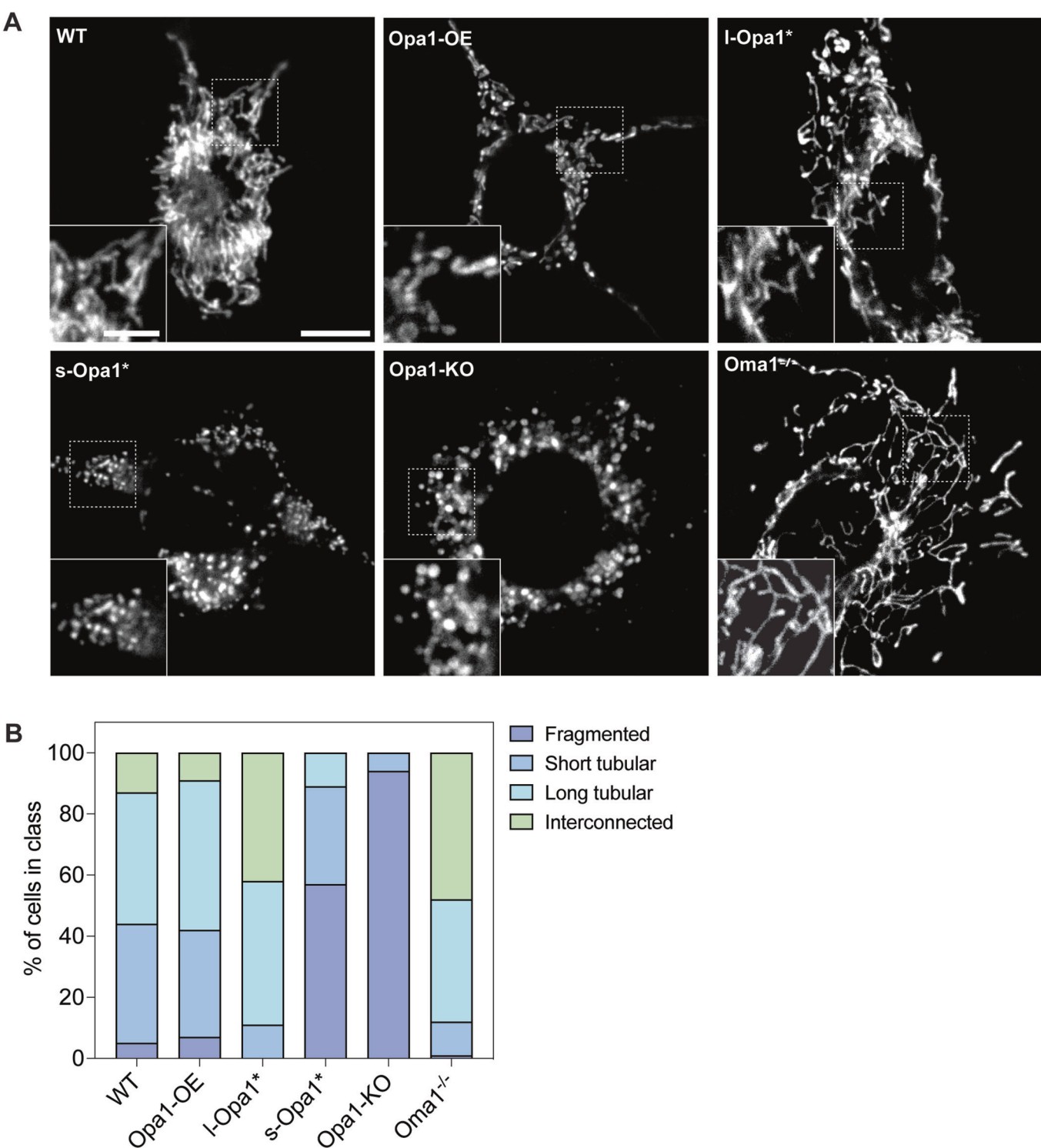

**Figure EV3.   Mitochondrial network morphology in MEF lines by fluorescence microscopy.**

(A) Representative images of mitochondrial morphology in indicated MEF lines labeled with MitoTracker™ Deep Red FM. Insets show magnified view of regions indicated with dashed boxes. Scale bar = 10 μm. Inset scale bar = 5 μm. (B) Graph bar representing mitochondrial network morphology scored in indicated MEF lines. $N = 100$ cells analyzed per cell line.

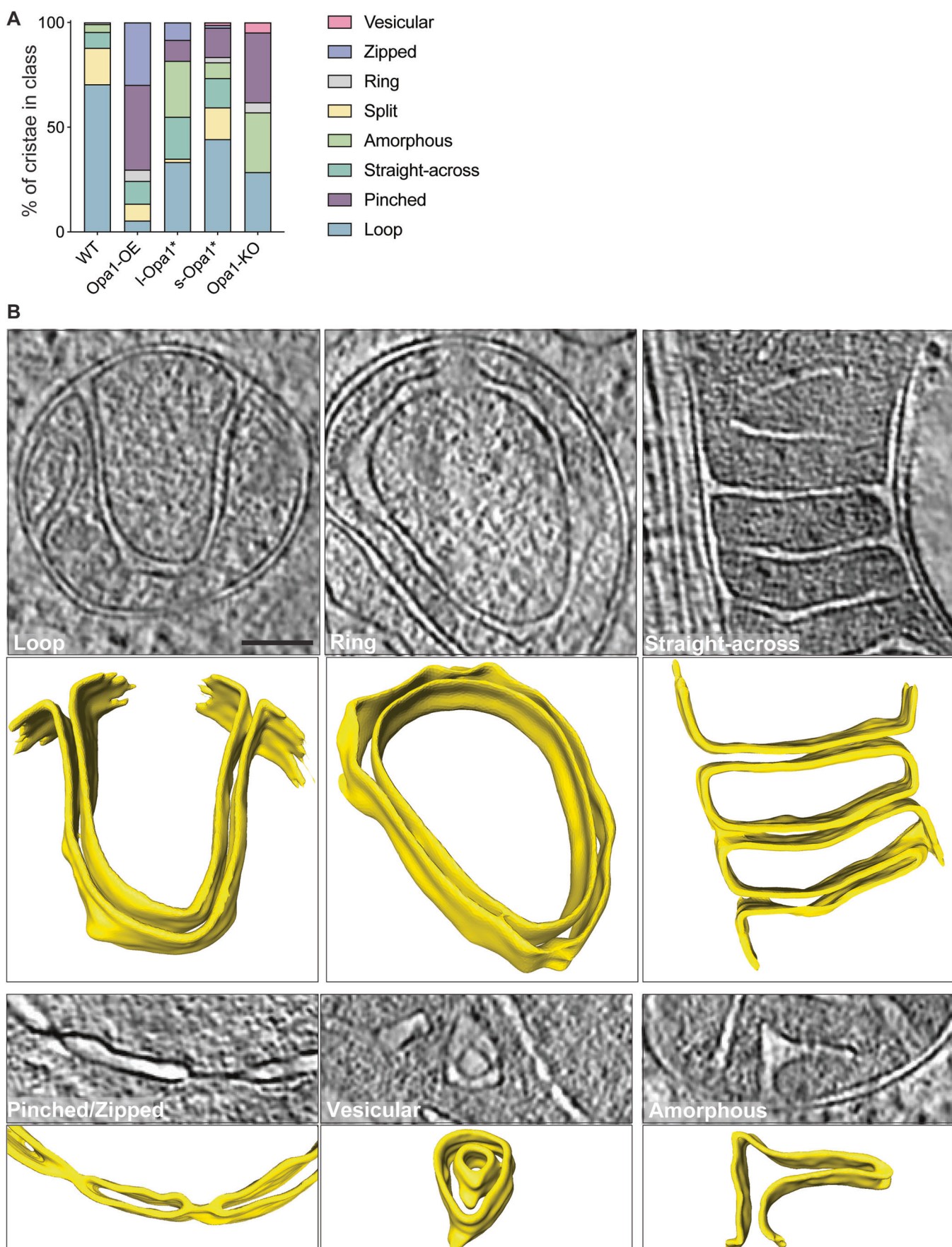

◀  **Figure EV4.  Unusual cristae morphology.**

(A) Graph bar representing the relative proportion of unusual cristae morphology observed in indicated MEF lines. Unusual cristae were categorized into vesicular, zipped, ring, split, amorphous, straight-across, pinched and loop. *N* refers to number of cristae analyzed, *N*: wild-type = 222, Opa1-OE = 430, l-Opa1* = 323, s-Opa1* = 653, Opa1-KO = 243. (B) Summed, projected central slices of cryo-electron tomograms showing examples of unusual cristae in mitochondria across cell lines in 2D (top) and 3D (bottom). Loop (from Fig. 1A, s-Opa1*), ring, straight-across (from Fig. 1A, l-Opa1*), pinched (from Appendix Fig. S2A, Opa1-KO second from the left), vesicular (from Fig. 1A, Opa1-KO), and amorphous cristae are shown. Scale bar = 200 nm.

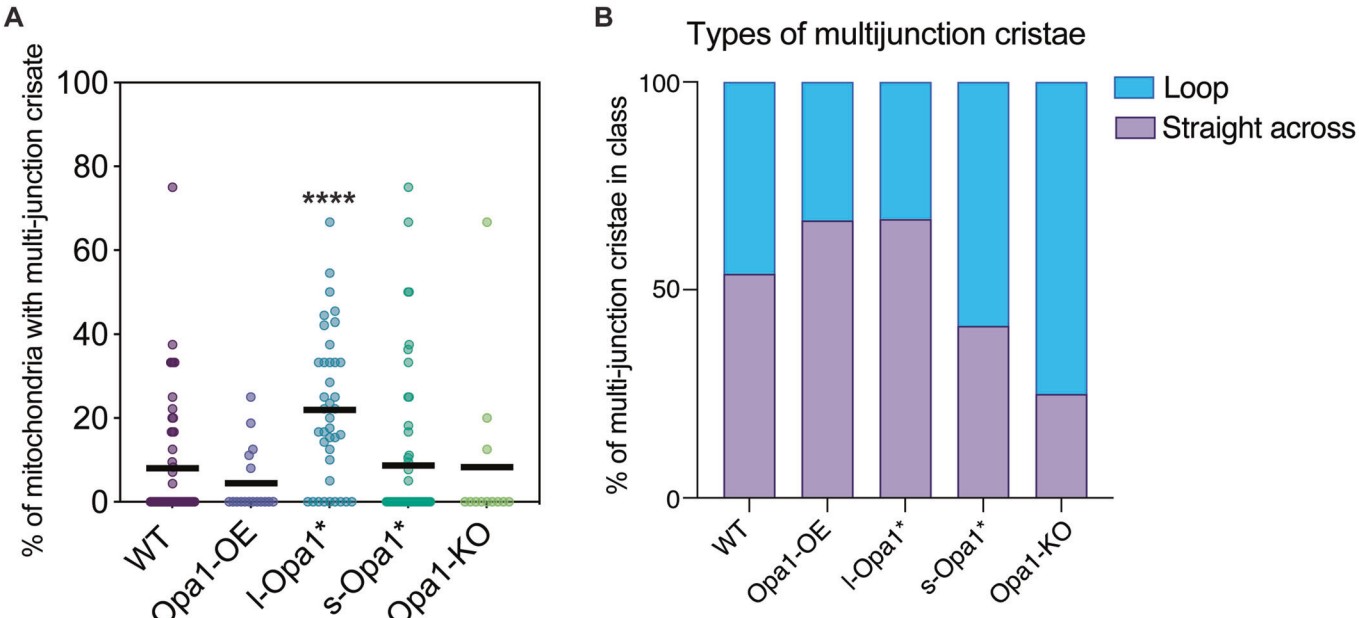

**Figure EV5.  Multijunction cristae.**

(A) Scatter plot showing the percentage of multijunction cristae per mitochondrion in indicated MEF lines. *N* refers to number of cristae; WT = 18, Opa1-OE = 5, l-Opa1* = 30, s-Opa1* = 16, Opa1-KO = 3. (B) Graph bar representing percentage of multijunction cristae categorized into straight-across and loop morphology in each MEF line. *N* refers to number of cristae; WT = 26, Opa1-OE = 9, l-Opa1* = 79, s-Opa1* = 29, Opa1-KO = 4. Data information: Scatter plot shows data distribution, the mean is marked by a bold black line. Significance of difference is tested relative to wild type using Mann–Whitney; ****p < 0.0001.

