## [Peer Review File · The EMBO Journal]

In situ architecture of Opa1-dependent mitochondrial cristae remodeling

Michelle Fry, Paula Navarro, Pusparanee Hakim, Virly Ananda, Xingping Qin, Juan Landoni, Sneha Rath, Zintes Inde, Camila Lugo, Bridget Luce, Yifan Ge, Julie McDonald, Ilzat Ali, Leillani Ha, Benjamin Kleinstiver, David Chan, Kristopher Sarosiek, and Luke Chao

DOI: [10.15252/embj.2023113495](https://doi.org/10.15252/embj.2023113495)

Corresponding author(s): Luke Chao (chao@molbio.mgh.harvard.edu)

Review Timeline:

Submission Date:	14th Jan 23
Editorial Decision:	24th Feb 23
Revision Received:	5th Oct 23
Editorial Decision:	16th Nov 23
Revision Received:	15th Dec 23
Accepted:	22nd Dec 23

Editor: William Teale

Transaction Report:

Dear Dr. Chao,

Thank you again for the submission of your manuscript entitled "In situ architecture of Opa1-dependent mitochondrial cristae remodeling" (EMBOJ-2023-113495) to our editorial office. We have now received reports from three referees, which I copy below.

As you can see from their comments, all referees appreciated the level of technical skill that you bring to the analysis of mitochondrial cristae structure. Furthermore, all referees agreed that your research question was well framed and the topic is exciting and timely. That said, referees 1 and 2 both pointed to a lack of mechanistic insight in your work. For example, your conclusions about the role s-Opa1 plays in shaping cristae, as reviewer 1 points out, need support from more direct functional experimentation. Reviewer 2 also makes a strong point that orthogonal direct functional tests are lacking. I agree that these tests are required, especially when your use of cryo-ET allows more established hypotheses to be re-appraised.

However, based on the overall interest expressed by the referees, I would like to invite you to address the comments of all referees in a revised version of the manuscript. I should add that it is The EMBO Journal policy to allow only a single major round of revision and that it is therefore important to resolve the main concerns, addressing the manuscript's lack of functional data, at this stage. I believe the concerns of the referees are reasonable and addressable; however, if you judge that the requested round of functional experimentation is not feasible please contact me, and we can decide together the best way forward. I would be happy to talk all this over via Zoom. Please, follow the instructions below when preparing your manuscript for resubmission.

I would also like to point out that as a matter of policy, competing manuscripts published during this period will not be taken into consideration in our assessment of the novelty presented by your study ("scooping" protection). We have extended this 'scooping protection policy' beyond the usual 3 month revision timeline to cover the period required for a full revision to address the essential experimental issues. Please contact me if you see a paper with related content published elsewhere to discuss the appropriate course of action.

Again, please contact me at any time during revision if you need any help or have further questions.

Thank you very much again for the opportunity to consider your work for publication. I look forward to your revision.

Best regards,

William Teale

William Teale, Ph.D.
Editor
The EMBO Journal

When submitting your revised manuscript, please carefully review the instructions below and include the following items:

- 1) a .docx formatted version of the manuscript text (including legends for main figures, EV figures and tables). Please make sure that the changes are highlighted to be clearly visible.
- 2) individual production quality figure files as .eps, .tif, .jpg (one file per figure).
- 3) a .docx formatted letter INCLUDING the reviewers' reports and your detailed point-by-point response to their comments. As part of the EMBO Press transparent editorial process, the point-by-point response is part of the Review Process File (RPF), which will be published alongside your paper.
- 4) a complete author checklist, which you can download from our author guidelines ([https://wol-prod-cdn.literatumonline.com/pb-assets/embo-site/Author Checklist%20-%20EMBO%20J-1561436015657.xlsx](https://wol-prod-cdn.literatumonline.com/pb-assets/embo-site/Author%20Checklist%20-%20EMBO%20J-1561436015657.xlsx)). Please insert information in the checklist that is also reflected in the manuscript. The completed author checklist will also be part of the RPF.
- 5) Please note that all corresponding authors are required to supply an ORCID ID for their name upon submission of a revised

manuscript.

6) We require a 'Data Availability' section after the Materials and Methods. Before submitting your revision, primary datasets produced in this study need to be deposited in an appropriate public database, and the accession numbers and database listed under 'Data Availability'. Please remember to provide a reviewer password if the datasets are not yet public (see <https://www.embopress.org/page/journal/14602075/authorguide#datadeposition>). If no data deposition in external databases is needed for this paper, please then state in this section: This study includes no data deposited in external repositories. Note that the Data Availability Section is restricted to new primary data that are part of this study.

Note - All links should resolve to a page where the data can be accessed.

7) When assembling figures, please refer to our figure preparation guideline in order to ensure proper formatting and readability in print as well as on screen:
<http://bit.ly/EMBOPressFigurePreparationGuideline>

8) For data quantification: please specify the name of the statistical test used to generate error bars and P values, the number (n) of independent experiments (specify technical or biological replicates) underlying each data point and the test used to calculate p-values in each figure legend. The figure legends should contain a basic description of n, P and the test applied. Graphs must include a description of the bars and the error bars (s.d., s.e.m.).

9) We would also encourage you to include the source data for figure panels that show essential data. Numerical data can be provided as individual .xls or .csv files (including a tab describing the data). For 'blots' or microscopy, uncropped images should be submitted (using a zip archive or a single pdf per main figure if multiple images need to be supplied for one panel). Additional information on source data and instruction on how to label the files are available at .

10) We replaced Supplementary Information with Expanded View (EV) Figures and Tables that are collapsible/expandable online (see examples in <https://www.embopress.org/doi/10.15252/embj.201695874>). A maximum of 5 EV Figures can be typeset. EV Figures should be cited as 'Figure EV1, Figure EV2' etc. in the text and their respective legends should be included in the main text after the legends of regular figures.

12) Our journal encourages inclusion of *data citations in the reference list* to directly cite datasets that were re-used and obtained from public databases. Data citations in the article text are distinct from normal bibliographical citations and should directly link to the database records from which the data can be accessed. In the main text, data citations are formatted as follows: "Data ref: Smith et al, 2001" or "Data ref: NCBI Sequence Read Archive PRJNA342805, 2017". In the Reference list, data citations must be labeled with "[DATASET]". A data reference must provide the database name, accession number/identifiers and a resolvable link to the landing page from which the data can be accessed at the end of the reference. Further instructions are available at .

Additional instructions for preparing your revised manuscript:

When assembling figures, please refer to our figure preparation guideline in order to ensure proper formatting and readability in

print as well as on screen:

We realize that it is difficult to revise to a specific deadline. In the interest of protecting the conceptual advance provided by the work, we recommend a revision within 3 months (25th May 2023). Please discuss the revision progress ahead of this time with the editor if you require more time to complete the revisions. Use the link below to submit your revision:

Referee #1:

Mitochondria change their shape and internal architecture and tailor their activities to adapt to cellular metabolic demands. Opa1 is a central player in these processes, regulating mitochondrial fusion, cristae morphogenesis and respiration. A long form of Opa1 can be cleaved by two peptidases, converting l-Opa1 into s-Opa1. The steady state levels of Opa1 forms are critical for mitochondrial morphology: while balanced accumulation of l- and s-Opa1 maintains normal morphologies, unbalanced steady state levels or overexpression impairs mitochondrial structure. Insights how Opa1 affects mitochondrial cristae resulted from cryoEM structures of Opa1/Mgm1 but important questions remained unaddressed (for instance, concerning the role of Opa1 processing). The current study aims at filling this gap, performing cryo-FIB of mitochondria in MEFs expressing different forms and levels of Opa1. Similar studies have been previously only performed by TEM in chemically fixed cells.

The strength of the study is the use of a cryo-ET technique that largely excludes possible artifacts due to fixation. There is no doubt that a detailed analysis of the role of Opa1 for cristae formation using cryo-FIB is of interest. Unfortunately, however, the novelty of the reported findings is limited. While some observations confirm earlier reports (for instance, wider cristae in the presence of s-Opa1* and increased cristae stacking in the presence of l-Opa1), others appear to contradict previous findings (see point 1). Although this might be of interest, further analysis is required to explain these apparent discrepancies. Moreover, I have major conceptual concerns about the used cell lines (point2), hampering the interpretation of the described data. The authors describe a plethora of different cristae shapes whose meaning and relevance remains unclear. The manuscript therefore remains very descriptive and provides only very limited new insight into the regulation of mitochondrial cristae by Opa1 and its different forms.

Specific points:

1. Two findings seemingly contradict previous reports. First, it has been proposed that Opa1-mediated tightening of cristae junctions restricts cytochrome c release from mitochondria protecting cells from apoptosis. In contrast, Opa1^{-/-} MEFs are shown here to be resistant to apoptosis, although mitochondrial cristae junctions are widened. Expression of l-Opa1, but not s-Opa1, tightened cristae junctions but sensitized cells for apoptotic insults. The authors proposed a direct role of l-Opa1 in facilitating cytC release during apoptosis, without providing experimental support for this hypothesis. Second, the authors suggest that s-Opa1 has limited activities in the maintenance of cristae shape. This somehow contradicts a recent cryo-EM structure of yeast Mgm1 and the previous finding that s-Opa1 (derived from Opa1-V5 in MEFs as in the present study) can replace endogenous Opa1 in cristae maintenance. Without additional functional evidence, the reader is left with these apparently disparate observations, making conclusions difficult. As a side note, the cristae defects observed in Opa1^{-/-} MEFs are milder than those

seen in TEM. Is there technical explanation for this notion?

2. The authors use Oma1^{-/-} deficient cells to study the effect of L-Opa1* and compare cristae shapes with WT cells. In my opinion, this is not correct, considering that Oma1 has additional functions (for instance regulating stress responses) that may indirectly affect mitochondrial morphologies. Not least, metabolic alterations are known to have profound effects on mitochondrial morphologies. Moreover, Oma1 was reported to bind to MICOS and play a role in the maintenance of cristae architecture. Related to this point, cells express different ratios of L- and S-Opa1 rather than exclusively only one form, which is why the authors refer to the cells as L-Opa1* and S-Opa1*, making interpretations on the functional role of both forms difficult if not impossible. Are the forms always assembled into Opa1 complexes? Moreover, what is the nature of the small band accumulating in L-Opa1* cells (Fig. S1)? These cells express an Opa1 variant lacking exon5b (i.e. lack the Yme1 cleavage site S2) and lack Oma1? Previous reports have demonstrated that exclusively L-Opa1 accumulates in MEFs lacking both processing peptidases. Fig. S1 would also require careful quantification considering the critical role of Opa1 levels for mitochondrial structure. The authors should also state clearly whether only stable cell lines were analyzed (rather than transiently transfected cell lines), to exclude different expression levels in different cells. It is a concern that cell lines originated from different sources or were established in different ways, but then used for comparative analysis.

Referee #2:

This is an elegant study that provides very high resolution images of cristae architecture in relationship to OPA1 variants.

While the manuscript provide ample descriptive information on the different cristae architecture parameters observed for the different OPA isoforms, there is a general lack of insight into the mechanism by which these OPA1 structural changes affect mitochondrial physiology.

Throughout the manuscript, the authors present a variety of architectural variations, yet it is not clear what are the functional consequences of these structural variations.

Each figure should report on the number of biological replicas (n=?) and the type of statistical analyses, and the p value. In addition, the definition of a biological replica should be stated. For example: are the n=4 meaning that the cells were transduced in 4 independent experiments run on different days? Is the number of cells analyzed used for the calculation of n= ? Which test was used to calculate p value?

In the BH3 profile, a decrease in cyto-c release is presented only for the OPA1 KO. Is this behavior also observed in OMA1 KO?

Is the processing of OPA not essential for the induction of apoptosis?

Authors show that L-OPA cells that lack OMA behave like WT cells. Is the conclusion that OPA1 processing is not necessary? If so, how cristae remodeling promote apoptosis? Authors should address this discrepancy from previous literature, cite the proper papers and provide their explanation for the opposing observations.

Another interesting point is the matrix density in the different cells. The manuscript does not provide an interpretation and discussion of the consequences of these findings. Is calcium buffering or sensitivity different for the cells? What is the correlation between cristae remodeling and matrix density? Is there a correlation between matrix density and apoptosis?

The model presented in the illustration is mostly based on the literature, not on the data of this manuscript. Readers that will look at this illustration will wonder what is new in this papers.

Referee #3:

Summary and Significance

The authors present a manuscript highlighting morphological comparisons of mitochondrial ultrastructure under different conditions of OPA1 expression and processing. The authors use the cellular cryo-electron tomography (cryo-ET) workflow to image cryo-focused ion beam (cryo-FIB) milled mouse embryonic fibroblasts expressing different Opa1 variants. They generate 3D segmentations of mitochondrial membranes visible in their tomograms and quantify several parameters of cristae architecture for comparative analyses across variants expressing different levels of s-Opa1 or l-Opa1. They show that Opa-1 regulates several aspects of mitochondrial cristae ultrastructure, including cristae/cristae junction length and width, volume, and matrix density. Through BH3 profiling, they demonstrate that these changes in cristae ultrastructure may be functionally linked to apoptotic priming in l-Opa1 cells. The tomograms and corresponding 3D segmentations are of exceptionally high quality, allowing for clear visualization of mitochondrial features. Additionally, the density-based analysis in several places helps to overcome potential biases from segmentations. Generally, the analyses are interesting and enhance our understanding of the

complex role of Opa-1 splicing in regulating mitochondrial cristae architecture its effect on mitochondrial apoptotic function.

Major Concerns:

1. Statistical analyses (general): Overall, there is a notable lack of details describing the statistical analyses performed, and the rationale for the selection of statistical tests used throughout the manuscript. More details are necessary in almost every case describing how the statistics being described are achieved. Since this paper centers on statistical assessment of mitochondrial morphology, more careful description is warranted. Determining the appropriate statistical tests for these types of analyses can be challenging, especially given the variation in membrane architecture observed within mitochondria in the same OPA1 variant group. The authors should include a section either in the main text and/or in the methods to describe the statistical tests performed and rationale for the selection of these statistical analyses. I encourage the authors to compare the results of different statistical tests to assess the robustness of the significance observed among the OPA1 variant groups. Additionally, it would be important to determine to what degree the major outliers observed in some of the quantifications (e.g., figs. 2C&D) are affecting the significance in the differences observed between these OPA1 variants and wild type.
2. Statistics analyses (cristae width): It is unclear how the statistics in Figure 2d come out of the cristae width measurement, based on a subtomogram extraction scheme. Is there a single point per crista, or a single point per subtomogram average? I am surprised to see multiple **** changes when the shift in mean seems quite small relative to the distribution of each violin. Why is the N=50 for all classes? Were a random subset of cristae or subvolumes chosen? What happens to these quantifications if a different box size is chosen? How does defocus affect the correlation to different templates?
3. Volume measurements: Several quantifications use area as a proxy for volume. This is probably a reasonable approximation, but it would be expected that roughly spherical objects (like mitochondria) would have a different proportional relationship between area as measured and true volume to roughly rectangular prismatic or cylindrical objects such as mitochondria cristae. Since figure 4 covers the true volume measurements, it would be helpful to understand how these metrics correlate for the sake of using area as a proxy for larger N measurements.
4. Inner-to-outer membrane quantification: There are several claims throughout the paper that may be better supported by measuring outer-to-inner membrane distance, rather than volume. For example, the authors note: "IMS volume appears similar in all cell lines except for s-Opa1* and Opa1-OE mitochondria, which both show larger relative IMS volume." I was surprised by this observation, since it appeared qualitatively in the exemplar images shown that the outer-to-inner membrane distance of OPA1-OE is reduced, which would likely lead to a lower IMS volume. Furthermore, the authors note, "Taken together, the absence of I-Opa1 results in larger IMS volumes, suggesting that I-Opa1 may play a role in maintaining WT OMM-IMM distances." Both instances point towards the importance of measuring inner-to-outer membrane distance as a better metric to quantify the changes observed across OPA1 variants. Therefore, I highly recommend that the authors consider making these inter-membrane measurements (or provide a justification for why they believe they are not relevant).
5. Relationship between network morphology and membrane ultrastructure: Given the essential roles of OPA1 in mediating not only membrane/cristae ultrastructure, but also mitochondrial network morphology via fusion-fission dynamics, it would be important to analyze the findings in the context of these two remodeling events. For example, are the differences in membrane ultrastructure observed in OPA1 variants due to the role in cristae remodeling, or due to biophysical constraints/flexibility imposed by modulating mitochondrial network morphology/fusion-fission dynamics? Ideally, this would be achieved by correlating, on a single cell level, the bulk mitochondrial network morphology (elongated, fragmented, etc.) to membrane ultrastructure. However, this would require a substantial amount of additional work that is likely beyond the scope of this manuscript. Alternatively, the authors could provide quantifications for the bulk mitochondrial network morphologies observed (for example, quantifications corresponding to images shown in Supplementary Figure S3) such that the ultrastructural differences observed could be correlated to perhaps the most abundant mitochondrial network morphology. To what degree is mitochondrial network and membrane ultrastructure remodeling coupled across the OPA1 variants? Additionally, since it is mentioned that the distributions in figure 1b can explain changes in figure 1b, it would be helpful to see the 1c area measures broken down by the classification in figure 1b. Are round mitochondria or polygonal mitochondria different in size between conditions, or is it just a factor of the proportions of those classes?
6. Analyze quantifications relative to cristae shape: Although the majority of cristae can be described as "lamellar", there is still some heterogeneity in cristae shape across the OPA1 variants (Figure 2a&b). However, the quantifications describing many of the various membrane architectures are done in bulk, presumably quantifying across all these different cristae shapes. It is not clear if the different cristae shapes represent different functional states and/or remodeling intermediates, and therefore it may be difficult to fully understand the contribution of the distinct OPA1 variants when analyzed in aggregate. To better understand the regulation of OPA1 variants in mediating cristae architecture, it would be useful to analyze these different membrane parameters (cristae length, width, junction width, junction angle, etc.) in the context of these different cristae shapes (straight, tilted, disconnected, lamellar, etc.). For example, do all lamellar cristae across OPA1 variants have similar cristae/junction widths, lengths, and angles? Or do certain shapes show substantial differences across OPA1 variants?
7. Classification of cristae ultrastructure (disconnected): For the disconnected classification, how confident are the authors that the cristae is truly disconnected? Given the fact that the cryo-FIB milling may inadvertently cut off portions of the mitochondria, is

it possible to unambiguously claim that these membranes are indeed disconnected from the rest of the inner membrane? Is it possible that they are connected at a portion of the IMM that is out of the field of view of the tomogram?

8. Classification of cristae ultrastructure (pinching): Figure 2e: in the segmentations the authors indicate pinching of membranes in the segmentations denoted by the blue triangle. However, when referencing the tomogram above it is unclear where the membrane "pinching" occurs in this particular 2D slices. How confident are the authors that the density spanning is indeed membrane and not protein tethers? It may be useful to include additional 2D tomographic slices of these pinching/zippering events to better illustrate this cristae ultrastructure. Perhaps choosing multiple different slices to better illustrate phenotype. One example is s-OPA1* straight: in this tomogram slice, it seems that the membranes are tightly spaced but are separated.

9. Comparison to previous work: Previous studies have examined the effect on the loss of OPA1 on cristae ultrastructure, revealing that loss of OPA1 can cause dramatic remodeling and even complete loss of cristae. While this present manuscript reveals some defects in cristae ultrastructure, the results seem comparatively less pronounced relative to observations from previous work. Do the authors agree? If so, it would be useful to include more of a discussion of the differences observed in this study relative to previous work.

10. Contribution of and compensation by other cristae remodeling proteins: In addition to OPA1, several other proteins and lipids are involved in modulating cristae ultrastructure. Is it possible that genetic loss of OPA1 or changes in OPA1 variant may be compensated for by the upregulation or change in expression/mitochondrial localization of other cristae shaping proteins? Have the authors (or previous work) assayed for compensation, perhaps biochemically?

11. Quantifying cristae spacing: One of the most striking qualitative phenotypes observed in the l-OPA1* variant is the presence of extensive cristae stacking in some of their OPA1 variants, suggesting that l-OPA1 may be playing a (somewhat unappreciated) role in maintaining the spacing between junctions. This is a very interesting finding but is difficult to assess based the quantifications presented in Supp Fig 5. This claim could be strengthened by directly quantifying and comparing the distances between cristae bodies across variants to test whether l-OPA1 may be playing a role in maintaining cristae stacking/ordering.

12. Apoptotic Assay: In line 321: "However, the importance of rearrangement and disassembly of Opa1 complexes during apoptosis initiation, to facilitate crista junction opening, as discussed above, demonstrates that Opa1 also has a direct role in facilitating the release of cytochrome c release when apoptosis is initiated. This view is consistent with our finding that complete knock-out of Opa1 impairs cytochrome c release and can be protective against apoptosis-inducing agents." I still don't quite follow the logic in this. How would wider junctions lead to higher apoptotic resistance? If we are following the model that cristae junction tightening inhibits cyt c release, this seems counterintuitive. Perhaps incorporating these aspects into the model would help (see below).

13. Model: Overall, the model presented is overly simplistic and does not adequately illustrate many of the exciting findings that were presented throughout the paper. As presented, the model does not substantially change the current view in the field, which I think is contradictory to what their data demonstrates, and I feel that the authors are selling themselves a bit short on their model. For example, it is missing any description of the findings related to: cristae volume, potential interaction of s-opa1 at junctions, and the importance of l-opa1* in mediating cristae stacking, etc. Additionally, the model is lacking any context relative to the defects observed in fission-fusion dynamics, and how these may be additionally coupled to differences in membrane ultrastructure described throughout the manuscript. The model also shows that s-Opa1 plays a role in sustaining narrow tubular cristae, however Figure 2 demonstrates that the majority of cristae in S-Opa1* exhibit lamellar cristae. Finally, one of the most exciting aspects of this paper is the attempt to link differences in cristae ultrastructure from OPA1 variants (i.e., form) to apoptotic priming (i.e., function). This aspect is not entirely explained within the model which, as presented, weakens the overall significance and impact of this work. Even if the authors are not entirely confident in the functional implications of the differences observed, it is worth attempting to illustrate their findings more effectively (and noting any uncertainties) to better convey how this study advances the field.

Minor Concerns:

1. For a more general audience, it would be useful to include a more detailed rationale as to why cryo-ET may provide more insight to the role of OPA1 in mediating cristae structure compared to previous studies using fixed or heavy metal-stained TEM methods.
2. How are the confidence intervals used when describing areas in figure 1c calculated? E.g. 0.34 ± 0.03 square microns for WT? The precision seems considerably lower when looking at the actual violin plot.
3. For cristae density: what criteria was used to determine what constituted an individual crista?
4. Starting in line 159: "Indeed, cells with altered levels and forms of Opa1 (Opa1-OE, l-Opa1* and s-Opa1* lines) have a reduced proportion of tubular and globular cristae, with a greater variation of cristae shape observed in s-Opa1* cells." It appears wild type has the greatest variation (and not s-Opa1*)? Could the authors clarify this point?
5. It would be useful to include a few more sentences in the beginning of the BH3 profiling section to provide a smoother transition and rationale between cristae shape sections and functional assays.
6. Figure S2 appears to be missing panels (d,e).

7. The meaning of the ****'s differs from figure to figure which is quite confusing to read.
8. Some of the classification was done by visual inspection as stated in the methods (1b,2b). If the authors could provide some information on how this classification was done. Was it an individual person who classified or were there a few people who classified based off of exemplar images?
9. In figure 2g the color rendering of the averages should correspond to the colors of the histograms (f) and quantifications (c,d).
10. How was targeting performed during tilt series acquisition? Was there bias in the targeting towards mitochondria, or mitochondria of a certain phenotype?
11. Starting in line 121: "Aligned with previous reports (Gómez-Valadés et al, 2021), mitochondria from Opa1-KO cells are larger than WT, but have a similar abundance and total mitochondrial area per cell as WT (Fig. S4d)." This is a bit confusing and seems contradictory. In the light microscopy data (Supp Figure 3), wild type has extensive elongation and Opa1-KO shows highly fragmented mitochondria. Could the authors clarify this point more clearly?
12. Despite the wild type and I-Opa1* cells showing hyperfused mitochondrial networks (Supp Figure 3), the mitochondria imaged by conventional TEM in Supp Figure 4 are highly fragmented. Could the authors explain the discrepancy between mitochondrial morphology observed between these data?
13. Starting in line 118: "Mitochondrial matrix density was classified as normal, dark, uneven or empty, which reflects brighter matrix staining (Fig. S4c). Matrix staining is prone to artifacts introduced by heterogenous heavy-metal stain and/or resin embedding." Could the authors explain the rationale in assaying the matrix texture using conventional TEM considering the described artifacts?

Non-essential suggestions for improving the study (which will be at the author's/editor's discretion)

1. Starting in line 326: "Mitochondrial shape and the curvature of the outer membrane has been previously found to affect the ability of the pro-apoptotic, pore-forming protein BAX to be stabilized in the mitochondrial outer membrane and initiate mitochondrial outer membrane permeabilization (MOMP) in cytochrome c release." Given the importance of curvature in mediating assembly of these essential pro-apoptotic proteins and the role of Opa-1-mediated apoptotic membrane remodeling, it may be quite interesting to examine how membrane curvature changes in the various Opa1 variants.

Dear Dr. Teale,

We thank the reviewers for their thoughtful comments and critiques, and are pleased to respond with a revised manuscript that we believe is much clearer and scientifically stronger.

This work is unique and noteworthy for the state-of-the-art technology and methodology employed, and its comprehensive nature, with an exceptional number of cell lines analyzed (5 different genetic conditions imaged by cryo-FIB/cryo-ET), which allow us to make new connections between cristae ultrastructure and function.

In our revised manuscript, we put forth a more comprehensive structural study on Opa1-dependent cristae remodeling. Beyond characterization of apoptotic responses, we now provide a suite of additional functional data linked to cristae morphologies, including mitochondrial calcium handling and respiration activity. Together, our structural data and these assays allows us to draw more definitive relationships between cristae architecture and mitochondrial function. We also expanded our fluorescence microscopy data and performed additional quantification to characterize cell mitochondrial network dynamics and mtDNA nucleoid distribution, in addition to a thorough characterization of the Opa1^{-/-} cell background. We have performed additional in-depth image analysis and exhaustive quantification of the structural data and made sure that all statistically relevant information has been well documented and noted.

We hope you will also find an improved discussion and summary figure highlighting our notable new findings from our *in situ* ultrastructural characterization of mitochondrial cristae with different forms of Opa1. This includes the observations that I-Opa1 expressing cells with WT-like cristae junction properties, show wild-type apoptotic response and calcium handling; and imbalance in Opa1 processing show compromised respiratory function and an increase in amorphous cristae.

We're grateful for the opportunity to address the points raised. Thank you again for your time and efforts with our manuscript.

Best wishes,

Luke Chao (on behalf of the co-authors)

Referee #1:

Mitochondria change their shape and internal architecture and tailor their activities to adapt to cellular metabolic demands. Opa1 is a central player in these processes, regulating mitochondrial fusion, cristae morphogenesis and respiration. A long form of Opa1 can be cleaved by two peptidases, converting l-Opa1 into s-Opa1. The steady state levels of Opa1 forms are critical for mitochondrial morphology: while balanced accumulation of l- and s-Opa1 maintains normal morphologies, unbalanced steady state levels or overexpression impairs mitochondrial structure. Insights how Opa1 affects mitochondrial cristae resulted from cryoEM structures of Opa1/Mgm1 but important questions remained unaddressed (for instance, concerning the role of Opa1 processing). The current study aims at filling this gap, performing cryo-FIB of mitochondria in MEFs expressing different forms and levels of Opa1. Similar studies have been previously only performed by TEM in chemically fixed cells.

The strength of the study is the use of a cryo-ET technique that largely excludes possible artifacts due to fixation. There is no doubt that a detailed analysis of the role of Opa1 for cristae formation using cryo-FIB is of interest. Unfortunately, however, the novelty of the reported findings is limited. While some observations confirm earlier reports (for instance, wider cristae in the presence of s-Opa1* and increased cristae stacking in the presence of l-Opa1), others appear to contradict previous findings (see point 1). Although this might be of interest, further analysis is required to explain these apparent discrepancies. Moreover, I have major conceptual concerns about the used cell lines (point2), hampering the interpretation of the described data. The authors describe a plethora of different cristae shapes whose meaning and relevance remains unclear.

The manuscript therefore remains very descriptive and provides only very limited new insight into the regulation of mitochondrial cristae by Opa1 and its different forms.

Specific points:

1. Two findings seemingly contradict previous reports. First, it has been proposed that Opa1-mediated tightening of cristae junctions restricts cytochrome c release from mitochondria protecting cells from apoptosis. In contrast, Opa1^{-/-} MEFs are shown here to be resistant to apoptosis, although mitochondrial cristae junctions are widened. Expression of l-Opa1, but not s-Opa1, tightened cristae junctions but sensitized cells for apoptotic insults. The authors proposed a direct role of l-Opa1 in facilitating cytC release during apoptosis, without providing experimental support for this hypothesis.

The reviewer is correct to point out that the widening of cristae junctions was linked to cytochrome C release in previous fixed/stained TEM work. In contrast to previous work, our observations of wider cristae junctions in s-Opa1* and KO-Opa1 cells are not in cells which have undergone apoptosis (after the addition of BH3-mimetic peptides), but rather in cells under normal, unperturbed conditions. In addition, our work is an improvement of previous studies, as the mitochondria were imaged by cryo-electron tomography meaning that the native structure of the sample is preserved and three-dimensional visualization is enabled.

We observed WT-like cristae junctions, and WT-like apoptotic cytochrome C release behavior in our l-Opa1* MEF cells (which were generated in an Oma1^{-/-} background). In contrast we observed that the s-Opa1* and Opa1-KO cells are not primed for cytochrome C release and have mitochondria with wider cristae junctions. **We emphasize that these**

observations are under steady-state conditions prior to any apoptotic stimuli. To also emphasize this clearly, we have mentioned it in the text (Lines 126-127, 208-210, 327-331).

The results from the apoptosis assay demonstrating that s-Opa1* and Opa1-KO cells are not primed for cytochrome C release, and that l-Opa1* cells respond like WT cells is our experimental evidence supporting the hypothesis that l-Opa1 plays a role in facilitating WT membrane transitions at cristae junctions.

We have adjusted our text and figures to clearly reflect and emphasize these new and surprising observations: Figure 4a, b; Figures S9; Lines 238-259, 336-346.

Second, the authors suggest that s-Opa1 has limited activities in the maintenance of cristae shape. This somehow contradicts a recent cryo-EM structure of yeast Mgm1 and the previous finding that s-Opa1 (derived from Opa1-V5 in MEFs as in the present study) can replace endogenous Opa1 in cristae maintenance.

We would like to emphasize that our results do not reflect on the *in situ* arrangement of Opa1, which, while a fascinating question, is outside the scope of this study. Several helical reconstructions of s-Opa1 (and its yeast orthologue s-Mgm1), including two recent reports, demonstrate helical assemblies (with more than one topology) (Faerber *et al.*, 2019, Zhang *et al.*, 2020, Nyenhuis *et al.*, 2023, Malsburg *et al.*, 2023). We cite and comment on their relevance with respect to tubular cristae in our discussion (Lines 314-317). It is also the case, that we do not observe any tubular cristae in our l-Opa1* condition. However, several pieces of evidence limit our ability to relate these cryo-EM and crystal structures with our data.

- a. Our l-Opa1* condition is mainly (>75% l-Opa1), but there still is some s-Opa1 present, which prevents us from concluding that s-Opa1 exclusively supports tubular cristae.
- b. There is considerable heterogeneity in the signal within the cristae lumen in our data, which prevents us from distinguishing any ordered protein assemblies.
- c. By definition, cryo-ET data is label-free and we cannot unambiguously identify the molecular identity of the densities within the cristae lumen.
- d. Previous cryo-EM/ET on Mgm1 studies were done *in vitro*.
- e. Previous cryo-EM studies of s-Opa1 were done *in vitro* by decorating the exterior surface of membrane tubes. Since tubular cristae have the opposite topology, it is unclear how these assemblies relate to tubular cristae. Rather, authors of recent papers propose that the tubulation observed in the s-Opa1 structures may facilitate fusion (Malsburg *et al.*, 2023).

In summary, we take care to not comment on the *in situ* arrangement of s-Opa1, as conclusive determination of its state will require other high-resolution structural approaches.

Second, with regards to the findings that Opa1-V5 in MEFs can replace endogenous Opa1, we again emphasize the difference between the TEM and cryo-ET imaging used in this study. Traditional TEM imaging of fixed stained images results in perturbations to native membrane shape. The harsh treatments of fixing, heavy metal staining and dehydration of

the sample, then embedding the sample in plastic resins results in sample swelling and introduction of great number of artifacts.

Without additional functional evidence, the reader is left with these apparently disparate observations, making conclusions difficult. As a side note, the cristae defects observed in Opa1^{-/-} MEFs are milder than those seen in TEM. Is there technical explanation for this notion?

In our revised manuscript, we have included a suite of new functional experiments selected because they are linked to cristae morphology. In addition to the BH3 profiling apoptotic priming assay (Figure 4a & b, Lines 228-259), we have now included, calcium handling measurements (Figure 4c & d, Lines 260-268), respiratory function measurements (Figure 5, Lines 269-277), and mtDNA distribution (Figure S11, Lines 286-296).

Our calcium measurements support our apoptosis observations, where I-Opa1 presence allows for WT-like responses to Ca²⁺ and apoptotic stimulation. Our oxygen consumption rate (OCR) measurements implicate amorphous cristae in compromised respiratory function.

Regarding the dramatic cristae defects observed in the Opa1^{-/-} MEF by TEM, please see again our comment above about perturbations caused by sample preparation artifacts for TEM imaging, as well as additional discussion in the text (Lines 54-55, 305-310). We again emphasize the quality of preservation of the native samples we are imaging by cryo-ET.

2. The authors use Oma1^{-/-} deficient cells to study the effect of L-Opa1* and compare cristae shapes with WT cells.

In my opinion, this is not correct, considering that Oma1 has additional functions (for instance regulating stress responses) that may indirectly affect mitochondrial morphologies. Not least, metabolic alterations are known to have profound effects on mitochondrial morphologies.

We acknowledge the alterations that may be due to the Oma1^{-/-} background, and as a result have secured the parent Oma1^{-/-} MEF cells in which the *Δexon5b* CRISPR line was generated in and undertaken additional functional assays to better characterize this background:

- a. We have also performed BH3 profiling for the Oma1^{-/-} background (Fig. S10a) and found intermediate rescue of WT BH3 profiling (Lines 279-280).
- b. We performed calcium retention capacity (CRC) assay on the Oma1^{-/-} MEF cells and show that this cell line has WT-like propensity for mPTP opening under Ca²⁺ bolus treatments but display a lower overall calcium buffering profile than WT and I-Opa1* cells (Fig. S10b, Lines 280-282).
- c. We measured OCR for the Oma1^{-/-} cells (Figure S10c-d), and found compromised basal and maximal respiratory capacity, compared to WT, and higher OCR levels to I-Opa1* cells. This demonstrates that while the Oma1^{-/-} background results in respiration defects, there are larger defects observed when the Opa1 levels are shifted to favor the long form (Lines 282-284).
- d.

In addition, several observations point to morphological similarities between I-Opa1* and WT conditions (which still have intact Oma1 activity). These include similar cristae junction

widths, cristae width, IMS volume and rare or no observations of globular and tubular cristae in both L-Opa1* and WT conditions. In addition, our observations of changes in the s-Opa1* condition (increase in tubular cristae, cristae junction width, cristae width, IMS volumes) are in a Oma1^{+/+} background.

Moreover, Oma1 was reported to bind to MICOS and play a role in the maintenance of cristae architecture.

We emphasize again the focus of our study: the effects to cristae morphology resulting from Opa1 forms. MICOS is certainly an important factor in cristae junction stability and maintenance, as a lack of MICOS results in the well-documented loss of CJs (Mukherjee et al., 2021). Our comparison of specific control conditions provide confidence in our comparisons of Opa1 form-dependent cristae effects. We note, once more, that in the Oma1-null *Δexon5b* CRISPR cells we observe WT-like CJ widths, and the changes to CJ widths we note are observed in KO and s-Opa1* cells where Oma1 has not been knocked-out. We also do not comment on cristae junction abundance in this study, as the cryo-ET data are only 150-200 nm-thick sections of mitochondria and not the entire mitochondria. We cannot conclude that cristae with no observed contact site are indeed disconnected when it is not possible to observe an entire continuous crista in our lamella. The role of Opa1 on cristae junction number is an interesting question outside the conceptual and technical scope of this work.

To assess the abundance of MICOS components in our cell lines of study, we performed a Western blot analysis against Mic60 and found its level to be similar across all cell lines investigated.

Related to this point, cells express different ratios of L- and S-Opa1 rather than exclusively only one form, which is why the authors refer to the cells as L-Opa1* and S-Opa1*, making interpretations on the functional role of both forms difficult if not impossible. Are the forms always assembled into Opa1 complexes?

The nature of the relevant Opa1 complexes important for cristae remodeling remain an active area of debate. We take care in this study to describe the cell conditions used in this study as containing mainly one form or the other and are careful not to over interpret. We again emphasize the structural cell biology nature of this study. The spatial distribution of

the different forms along the mitochondrial inner-membrane remains an interesting question for future study.

Moreover, what is the nature of the small band accumulating in L-Opa1* cells (Fig. S1)? These cells express an Opa1 variant lacking exon5b (i.e. lack the Yme1l cleavage site S2) and lack Oma1?

The short band in the l-Opa1* lane is a small fraction of the isoform 5 Opa1 expressed and processed into the short form by YME1L in a cleavage-site 2 (S2)-independent manner (*Wang et al. 2020; PMID 33237841*). The cell line does lack Oma1 and has a Yme1L S2 cleavage site removed (exon5b deletion) in order to preserve necessary mitochondria processing while still significantly reducing the processing of Opa1. We quantified the relative amount of s-Opa1 in the l-Opa1* cell line as ~23% out of all the Opa1 signal present.

Previous reports have demonstrated that exclusively L-Opa1 accumulates in MEFs lacking both processing peptidases. Fig. S1 would also require careful quantification considering the critical role of Opa1 levels for mitochondrial structure.

The Opa1 signal for all cell lines were normalized as demonstrated by the blot against β -actin (Fig S1a).

The authors should also state clearly whether only stable cell lines were analyzed (rather than transiently transfected cell lines), to exclude different expression levels in different cells. It is a concern that cell lines originated from different sources or were established in different ways, but then used for comparative analysis.

As discussed in our methods section (Lines 628-630), these cell lines are stable lines and have been used in previously published work (*Wang et al., 2021; PMID 33237841*). We have outlined the cell line backgrounds in the Methods section.

Referee #2:

This is an elegant study that provides very high resolution images of cristae architecture in relationship to OPA1 variants.

While the manuscript provide ample descriptive information on the different cristae architecture parameters observed for the different OPA isoforms, there is a general lack of insight into the mechanism by which these OPA1 structural changes affect mitochondrial physiology.

Throughout the manuscript, the authors present a variety of architectural variations, yet it is not clear what are the functional consequences of these structural variations.

We thank the reviewer for their encouraging words. We are glad to see agreement that our manuscript describes in detail cristae morphologies linked to different Opa1 conditions in cells using cryo-ET. This cutting-edge technique allows us to describe in 3D under native conditions what has only been described by 2D images, as well as compare between cells with only one majority form of Opa1. In our revised manuscript, we take the study one step further, by performing several additional functional assays (in addition to the apoptotic sensitivity assay (Fig. 4a-b)), to help relate cristae morphology with function. We performed respiration measurements (Fig. 5), calcium uptake assays (Fig 4c-d), and mtDNA distribution imaging and quantification (Fig. S11) because these functions are closely linked with cristae morphology.

In summary, we found notable relationships between mitochondrial function and cristae morphology (Fig 6):

- I-Opa1 expressing cells with WT-like cristae junction properties, show wild-type apoptotic response and calcium handling.
- Abnormal apoptotic and calcium responses seen in s-Opa1* cells with wider cristae junction and the presence of vesicular cristae.
- Imbalance in Opa1 processing show compromised respiratory function and an increase in amorphous cristae.

Each figure should report on the number of biological replicas (n=?) and the type of statistical analyses, and the p value. In addition, the definition of a biological replica should be stated. For example: are the n=4 meaning that the cells were transduced in 4 independent experiments run on different days? Is the number of cells analyzed used for the calculation of n= ? Which test was used to calculate p value?

We agree with the reviewer that these are critical details that should accompany all figures with statistical analyses. For that reason, these details were included in the figure legends in the original manuscript. The definitions of biological replicates are detailed in the Methods section (Lines 858-859, 885-887, 898-899, 928-930). We have now also included biological replicate numbers in all figure legends for further clarity.

In the BH3 profile, a decrease in cyto-c release is presented only for the OPA1 KO. Is this behavior also observed in OMA1 KO?

We measured the BH3 response profile for Oma1^{-/-} cells and found an intermediate restoration of apoptotic priming. This is consistent with an imbalance of I-Opa1:s-Opa1, in

favor of s-Opa1 correlating with a compromised apoptotic response (Figure S10a; Lines 279-280).

Is the processing of OPA1 not essential for the induction of apoptosis?

Our data suggests that the processing of Opa1 is not essential for WT-like response to apoptosis initiation as l-Opa1* cells responded to apoptosis initiation in a manner similar to WT. We have adjusted our text to specifically state this major new finding from our results (Lines 257-259; 344-346).

Authors show that L-OPA cells that lack OMA behave like WT cells. Is the conclusion that OPA1 processing is not necessary? If so, how cristae remodeling promote apoptosis? Authors should address this discrepancy from previous literature, cite the proper papers and provide their explanation for the opposing observations.

We detail and explain our conclusion that Opa1 *processing* is not necessary for apoptosis in our discussion, putting our work in context with previous literature. Based on our results, we reconcile previous findings by distinguishing multiple roles for Opa1 during apoptotic membrane transitions (Lines 336-346).

There is a cytochrome C retention role, as observed previously in knock-down cells which show increased cytochrome C release (*Olichon et al. 2002; PMID 12509422*) or apoptotic death (*Dong et al. 2021; PMID 34621753*). Consistent with this role, Opa1 overexpression reduced apoptotic cell death (*Dong et al. 2021, Varanita et al. 2015; PMID 26039448*).

From our measurements, we propose a rearrangement role to facilitate cristae junction changes. This step was suggested in previous work by Frezza et al. (*PMID 16839885*), which showed fast cytochrome C release in mitochondria isolated from GTPase dead and KD Opa1 cells, and slower cytochrome C release in mitochondria from Opa1-OE cells.

Another interesting point is the matrix density in the different cells. The manuscript does not provide an interpretation and discussion of the consequences of these findings. Is calcium buffering or sensitivity different for the cells?

This is an interesting suggestion. To address this, we have performed a calcium uptake assay (see Methods Lines: 891-899, Figure 4c-d). We found l-Opa1 presence allows for WT-like Ca²⁺ responses. Opa1-KO or s-Opa1* cells required greater CaCl₂ stimulation to undergo mPTP opening meanwhile Opa1-OE cells had a lower CaCl₂ response threshold.

We do not observe any correlation between matrix density and calcium handling in these cell lines. We note a less condensed matrix in s-Opa1* mitochondria and a lighter matrix in Opa1-KO mitochondria (Fig. S3g). This suggests matrix density differences are most likely attributable to calcium independent factors. We did observe, however, increased calcium phosphate deposits in Opa1-OE cell lines (Fig. S3h). We discuss this in lines: 351-353.

What is the correlation between cristae remodeling and matrix density?

From the work we presented in our first submission, we see that narrower cristae widths correlate with lower matrix density. The mitochondria in s-Opa1* cell lines have significantly narrower cristae and significantly lighter matrix density than WT mitochondria. We have adjusted the text to explicitly state this (Lines 355-357).

Is there a correlation between matrix density and apoptosis?

No, there is not a direct correlation between matrix density and apoptosis. Both Opa1-KO and s-Opa1* cells are resistant to apoptosis, but only s-Opa1* mitochondria have less dense matrices than WT. Opa1-KO mitochondria, in fact, have denser matrices than WT.

We do note a correlation between calcium retention capacity (Fig. 4c-d) and the apoptotic response (Fig. 4a-b) of these cell lines. l-Opa1* MEF have similar calcium response profile as WT (consistent with BH3 responses). s-Opa1* and Opa1-KO MEF are more resistant to mPTP opening (characterized by higher concentrations of CaCl₂ needed to induce mPTP opening).

The model presented in the illustration is mostly based on the literature, not on the data of this manuscript. Readers that will look at this illustration will wonder what is new in this papers.

We (and Reviewer 3) respectfully disagree. Our model reflects new insights from this work, which were not described in previous work in the literature. As Reviewer 3 correctly points out, many of the new observations from this work *contrast* with the current literature. We have revised the manuscript to highlight the technical novelty of this study (applying cryo-ET of cryo-FIB-milled lamellae) coupled to new functional insights.

In summary, we report:

1. First *in situ* cryo-ET description of how Opa1 form (long or short) shapes native membrane ultrastructure
2. New findings that the long form of Opa1 is important for mediating cytochrome C release and maintaining cristae junction width.
3. A completely processed form of s-Opa1 is insufficient to promote cytochrome C release and correlates with wider CJs under steady-state conditions.
4. New observation that the long form of Opa1 is important for WT-like calcium handling, s-Opa1* (and Opa1-KO) MEF are more resistant to mPTP opening, and Opa1-OE MEFs show reduced calcium buffering threshold.
5. New finding that l-Opa1* and s-Opa1* cells have defects in respiration tied to amorphous cristae.

We have taken Reviewer 3's comments into consideration and adjusted our model (Fig. 6) to accurately reflect and convey the novel discoveries our application of cryo-FIB and cryo-ET allowed us to make. We also speculate that presence of straight-across multijunction cristae in l-Opa1* cells may result from accumulation of fusion intermediates. We have restructured the discussion to incorporate the updated model (Fig. 6) and added clarifying text to help guide readers through these new findings.

Referee #3:

Summary and Significance

The authors present a manuscript highlighting morphological comparisons of mitochondrial ultrastructure under different conditions of OPA1 expression and processing. The authors use the cellular cryo-electron tomography (cryo-ET) workflow to image cryo-focused ion beam (cryo-FIB) milled mouse embryonic fibroblasts expressing different Opa1 variants. They generate 3D segmentations of mitochondrial membranes visible in their tomograms and quantify several parameters of cristae architecture for comparative analyses across variants expressing different levels of s-Opa1 or l-Opa1. They show that Opa-1 regulates several aspects of mitochondrial cristae ultrastructure, including cristae/cristae junction length and width, volume, and matrix density. Through BH3 profiling, they demonstrate that these changes in cristae ultrastructure may be functionally linked to apoptotic priming in l-Opa1 cells. The tomograms and corresponding 3D segmentations are of exceptionally high quality, allowing for clear visualization of mitochondrial features. Additionally, the density-based analysis in several places helps to overcome potential biases from segmentations. Generally, the analyses are interesting and enhance our understanding of the complex role of Opa-1 splicing in regulating mitochondrial cristae architecture its effect on mitochondrial apoptotic function.

We thank the reviewer for these encouraging words and the acknowledgement of the quality of our tomograms, analysis, and the impact of our work.

Major Concerns:

1. Statistical analyses (general): Overall, there is a notable lack of details describing the statistical analyses performed, and the rationale for the selection of statistical tests used throughout the manuscript. More details are necessary in almost every case describing how the statistics being described are achieved. Since this paper centers on statistical assessment of mitochondrial morphology, more careful description is warranted. Determining the appropriate statistical tests for these types of analyses can be challenging, especially given the variation in membrane architecture observed within mitochondria in the same OPA1 variant group. The authors should include a section either in the main text and/or in the methods to describe the statistical tests performed and rationale for the selection of these statistical analyses. I encourage the authors to compare the results of different statistical tests to assess the robustness of the significance observed among the OPA1 variant groups. Additionally, it would be important to determine to what degree the major outliers observed in some of the quantifications (e.g., figs. 2C&D) are affecting the significance in the differences observed between these OPA1 variants and wild type.

We have adjusted our Methods section to reflect the reasoning specific statistical tests were used (Lines 837-844, 859-864, 887-889, 920-922, 937-939, 963-965)). We performed multiple statistical tests, all of which resulted in trends of similar significance, reflecting the robustness of our determination of significant differences. These are all now included in the source data. We have also addressed the outliers. Removal of these outliers did not affect the significance of the differences reported. Please see, attached graphs that correspond to Fig. 2c and d replotted without outliers.

2. Statistics analyses (cristae width): It is unclear how the statistics in Figure 2d come out of the cristae width measurement, based on a subtomogram extraction scheme. Is there a single point per crista, or a single point per subtomogram average? I am surprised to see multiple **** changes when the shift in mean seems quite small relative to the distribution of each violin.

We measured cristae width per subtomogram particle, so there is a single point per particle and each particle is a segment of the cristae. The number of particles per cristae depend on the length of cristae. Particles are all cropped with the same interdistance and box size. We have clarified this in lines 817-830 and Fig. S7b. Regarding the violin plots, the bold line indicates median \pm quartiles and not mean \pm SD. To avoid confusion and provide a better representation of the raw data, we have changed all our plots to scattered plots. We have also indicated mean \pm SD for Fig. 2d in the legend (Lines 967-968).

Why is the N=50 for all classes? Were a random subset of cristae or subvolumes chosen?

We thank the reviewer for catching this error. There were several errors in Figure 2 legend regarding N numbers. We classified all cristae in all our tomograms, but measured 3D length for a random subset of 50 cristae per cell line. From those 50 cristae we picked particles for STA analysis and cristae width measurements, thus, number of particles for STA depends on cristae length. Particles were cropped with the same interdistance and box size.

What happens to these quantifications if a different box size is chosen?

The reviewer raises a fair point. We carefully chose the optimal box size by physically measuring intra cristae membranes distances. This distance was 38 pixels in our tomograms, we allowed 10 extra pixels so that cristae fit comfortably inside the physical box. This makes a total of 48 pixels.

Image showing assessment of box size in pixels in our tomograms.

To answer this question, we re-cropped particles using a tighter box size (36 pixels) and a larger box size (56 pixels). Our results show that our original box size is the most adequate since outliers are minimal or nonexistent. Increased number of outliers are probably due to tight edges close to cristae membrane densities. Also, for an overlarge box size (56 pixels), neighboring cristae densities might be present, misleading alignment. When outliers are removed, mean and SD are rescued and similar to the original box size (48 pixels).

Figure showing plots of cristae width raw data (top row) and without outliers (bottom row). Mean and standard deviation are indicated on top of each group which are: original (48) = box size of 48 pixels in XYZ; 36 = box size of 36 pixels in XYZ and 56 = box size of 56 in XYZ. N is stated in red on the bottom row and is the same for above plots. Number of outliers in each data set is stated in red on top of each group (bottom row plots).

We used the same parameters for all cell lines tested. The MATLAB script run under *Dynamo* environment is now available in Github:

(<https://github.com/NavarroPP/membraneThickness/blob/main/CristaeThickness.m>).

How does defocus affect the correlation to different templates?

We understand the point raised by Reviewer 3 on the contribution of defocus to template bias. This is relevant and important for structure determination, especially for high-resolution maps obtained using template matching for particle picking and/or use of a template for particle alignment. Our STA strategy does not aim to claim the determination of a structure and we do not claim any resolution, as our pixel size for this analysis is 2.2 nm. Further, we picked particles manually based on cristae morphology and we used a random average outputted from our particles as the initial template (Lines 817-830). As we only used templates to measure membrane-to-membrane distance, but not for picking or aligning particles, we do not expect template bias.

For all these reasons, especially the pixel size, and the fact that we are analyzing membranes which have a strong and unequivocal contrast, we believe that changes in defocus do not have an impact in our results. Furthermore, we have already tested the parameter box size, which did not affect our results and conclusions (see answer above).

3. Volume measurements: Several quantifications use area as a proxy for volume. This is probably a reasonable approximation, but it would be expected that roughly spherical objects (like mitochondria) would have a different proportional relationship between area as measured and true volume to roughly rectangular prismatic or cylindrical objects such as mitochondria cristae. Since figure 4 covers the true volume measurements, it would be helpful to understand how these metrics correlate for the sake of using area as a proxy for larger N measurements.

To address these concerns, we plotted 3D total volume (nm^3) against corresponding 2D area (nm^2). Overall, data shows proportionality, except for s-Opa1*. We suspect this might be due to the disparate mitochondria morphologies found in s-Opa1*.

Figure plotting mitochondrial 3D measurement (nm^3) against corresponding 2D measurements (nm^2) per cell line.

As the reviewer points out, previous Figure 4, now Figure S3 covers 3D measurements. The motivation of Figure S3 is to compare mitochondrial sub-compartments and not whole

mitochondria volumes. Any conclusion regarding whole mitochondrial volume might be misleading since our cryo-electron tomograms only cover a slab of the mitochondria, not the entire mitochondrial volume. Therefore, we do not make any claim regarding whole mitochondria 3D volume. Thus, we use 2D area as a better representation of mitochondrial size for two main reasons: (1) N numbers are higher and (2) previous conventional TEM data follow this strategy and therefore, we could make meaningful comparison between these data.

4. Inner-to-outer membrane quantification: There are several claims throughout the paper that may be better supported by measuring outer-to-inner membrane distance, rather than volume. For example, the authors note: "IMS volume appears similar in all cell lines except for s-Opa1* and Opa1-OE mitochondria, which both show larger relative IMS volume." I was surprised by this observation, since it appeared qualitatively in the exemplar images shown that the outer-to-inner membrane distance of OPA1-OE is reduced, which would likely lead to a lower IMS volume. Furthermore, the authors note, "Taken together, the absence of I-Opa1 results in larger IMS volumes, suggesting that I-Opa1 may play a role in maintaining WT OMM-IMM distances." Both instances point towards the importance of measuring inner-to-outer membrane distance as a better metric to quantify the changes observed across OPA1 variants. Therefore, I highly recommend that the authors consider making these inter-membrane measurements (or provide a justification for why they believe they are not relevant).

As the reviewer suggests we have made direct measurements of OMM and IMM distances. Given the challenges and caveats in comparing partial volumes of mitochondria, we have decided to exclude this analysis from the manuscript. We have performed the measurements and describe them here, below:

Figure IMM-OMM distances. N = 300 for all cell lines. Mean +/- SD: WT = 16.43 +/- 2.161; Opa1-OE = 12.63 +/- 3.324; I-Opa1* = 13.18 +/- 2.511; s-Opa1* = 13.78 +/- 1.908; Opa1-KO = 17.72 +/- 4.420. WT vs Opa1-OE: ****p<0.0001; WT vs I-Opa1*: ****p<0.0001; WT vs s-Opa1*: ****p<0.0001; WT vs Opa1-KO: *p<0.05.

5. Relationship between network morphology and membrane ultrastructure: Given the essential roles of OPA1 in mediating not only membrane/cristae ultrastructure, but also mitochondrial network morphology via fusion-fission dynamics, it would be important to analyze the findings in

the context of these two remodeling events. For example, are the differences in membrane ultrastructure observed in OPA1 variants due to the role in cristae remodeling, or due to biophysical constraints/flexibility imposed by modulating mitochondrial network morphology/fusion-fission dynamics? Ideally, this would be achieved by correlating, on a single cell level, the bulk mitochondrial network morphology (elongated, fragmented, etc.) to membrane ultrastructure. However, this would require a substantial amount of additional work that is likely beyond the scope of this manuscript. Alternatively, the authors could provide quantifications for the bulk mitochondrial network morphologies observed (for example, quantifications corresponding to images shown in Supplementary Figure S3) such that the ultrastructural differences observed could be correlated to perhaps the most abundant mitochondrial network morphology.

We thank the reviewer for their suggestion. We have expanded our fluorescent microscopy data to include quantification of bulk mitochondrial network morphologies to better correlate with ultrastructural differences observed in the cell lines via cryo-ET (Fig. S5b). Mitochondria network morphologies were quantified and classified on a per cell basis as done previously (Wang et al., 2021 PMID: 33237841).

To what degree is mitochondrial network and membrane ultrastructure remodeling coupled across the OPA1 variants? Additionally, since it is mentioned that the distributions in figure 1b can explain changes in figure 1c, it would be helpful to see the 1c area measures broken down by the classification in figure 1b. Are round mitochondria or polygonal mitochondria different in size between conditions, or is it just a factor of the proportions of those classes?

We have extended our classifications to breakdown mitochondrial size by mitochondria shape classification. This is now included in Fig. S2b. Indeed, between conditions, round or polygonal mitochondria are different in size.

6. Analyze quantifications relative to cristae shape: Although the majority of cristae can be described as "lamellar", there is still some heterogeneity in cristae shape across the OPA1 variants (Figure 2a&b). However, the quantifications describing many of the various membrane architectures are done in bulk, presumably quantifying across all these different cristae shapes. It is not clear if the different cristae shapes represent different functional states and/or remodeling intermediates, and therefore it may be difficult to fully understand the contribution of the distinct OPA1 variants when analyzed in aggregate. To better understand the regulation of OPA1 variants in mediating cristae architecture, it would be useful to analyze these different membrane parameters (cristae length, width, junction width, junction angle, etc.) in the context of these different cristae shapes (straight, tilted, disconnected, lamellar, etc.). For example, do all lamellar cristae across OPA1 variants have similar cristae/junction widths, lengths, and angles? Or do certain shapes show substantial differences across OPA1 variants?

We respectfully disagree. We believe that the suggested breakdown will not help distinguish general traits, but may over-interpret the effect of Opa1 variants as some of the classes are very small in number. Furthermore, we do not claim any sequence or intermediate states in cristae remodeling. We have already shown before that breakdown classes (qualitative) by physical parameters (quantitative), i.e., size, do not show any new trends. However, the data show that the Opa1 condition (WT, Opa1-OE, l-Opa1*, s-Opa1* and Opa1-KO) is the factor influencing the resulting measurements (see point 3 and Fig. S3, regarding subcompartment volume).

7. Classification of cristae ultrastructure (disconnected): For the disconnected classification, how

confident are the authors that the cristae is truly disconnected? Given the fact that the cryo-FIB milling may inadvertently cut off portions of the mitochondria, is it possible to unambiguously claim that these membranes are indeed disconnected from the rest of the inner membrane? Is it possible that they are connected at a portion of the IMM that is out of the field of view of the tomogram?

We classified cristae as disconnected based on the lack of a visible attachment to the IMM, as described in lines 151-152 “For some cristae no connection to the IMM was captured in the tomogram and thus classified as no attachment observed (NAO)”. As the reviewer points out, it is possible that the attachment is not captured in our tomograms and this is what we had hoped to convey in our category label (disconnected). We have changed this category to no attachment observed (NAO) in Fig. 2.

8. Classification of cristae ultrastructure (pinching): Figure 2e: in the segmentations the authors indicate pinching of membranes in the segmentations denoted by the blue triangle. However, when referencing the tomogram above it is unclear where the membrane "pinching" occurs in this particular 2D slices. How confident are the authors that the density spanning is indeed membrane and not protein tethers? It may be useful to include additional 2D tomographic slices of these pinching/zippering events to better illustrate this cristae ultrastructure. Perhaps choosing multiple different slices to better illustrate phenotype. One example is s-OPA1* straight: in this tomogram slice, it seems that the membranes are tightly spaced but are separated.

We had attempted to convey that this pinching and zippering was throughout a 3D space with our 3D renderings. This pinching or zippering is clearly shown through the z-axis in Fig. S6, and also in the supplementary movies, as cristae shape is best displayed in tomograms and 3D renderings. We found that using pinched or zipped in Fig. 2e might be confusing since sometimes membranes pinch to split cristae or detach from cristae junctions at specific points but not throughout all the volume. We have removed those labels to avoid confusion. Here we show slices of the 3D for s-Opa1* in Fig. 2e as an example:

9. Comparison to previous work: Previous studies have examined the effect on the loss of OPA1 on cristae ultrastructure, revealing that loss of OPA1 can cause dramatic remodeling and even complete loss of cristae. While this present manuscript reveals some defects in cristae ultrastructure, the results seem comparatively less pronounced relative to observations from previous work. Do the authors agree? If so, it would be useful to include more of a discussion of the differences observed in this study relative to previous work.

Like previous reports, we also see an increase in globular and heterogeneous cristae morphologies (Fig. 2). To our knowledge the loss of Opa1 results in cristae morphology

heterogeneity and a reduction in number of cristae but not a loss of cristae (PMID: 28298442; PMID: 12509422; PMID: 25298396; PMID: 23138851). These studies were done with fixed samples using traditional transmission electron microscopy. Because our aim was to capture structurally well-preserved data, our field of view was limited and we could only image one or at most a few mitochondria at a time. In order to optimize our resources, targets were selected from screening images of low contrast because we were conscious of the amount of dosage we exposed our sample to before collection to limit radiation damage. A consequence of this was that we were searching for structures that were clearly mitochondria, i.e. double membrane organelles with invaginations in the inner membrane. It was notably more difficult to select targets in Opa1-KO MEFs than in other cell lines.

10. Contribution of and compensation by other cristae remodeling proteins: In addition to OPA1, several other proteins and lipids are involved in modulating cristae ultrastructure. Is it possible that genetic loss of OPA1 or changes in OPA1 variant may be compensated for by the upregulation or change in expression/mitochondrial localization of other cristae shaping proteins? Have the authors (or previous work) assayed for compensation, perhaps biochemically?

There are three other known regulators of cristae shape Opa1, Complex V, and the MICOS complex. Previous studies have knocked out or down each of these regulators individually and observed changes to cristae morphology (PMID: 34847778; PMID: 11823415; PMID: 19528297; PMID: 27849155; PMID: 12808034). These changes appear to be independent of each other. Opa1-KO results in fewer and more heterogeneous cristae morphologies. Knocking down subunits of Complex V results in the reduction of cristae tips. And knocking down subunits of MICOS resulted in the elimination of cristae junctions. Each of these changes appear to affect different regions of the cristae. In our study, we do not see a clear reduction of cristae junctions, suggesting the absence of each Opa1 form does not affect the regulation of cristae number by MICOS. While we see more multi-junction cristae in our I-Opa1* cell lines, this could also be attributed to stalled fusion intermediates (discussed in Lines 320-321; Fig 6c) and not an effect on Complex V regulation of cristae shape. While our study compares effects to cristae morphology by the two forms of Opa1, we cannot rule out that some changes may be due to indirect effects of altered Opa1 processing.

11. Quantifying cristae spacing: One of the most striking qualitative phenotypes observed in the I-OPA1* variant is the presence of extensive cristae stacking in some of their OPA1 variants, suggesting that I-OPA1 may be playing a (somewhat unappreciated) role in maintaining the spacing between junctions. This is a very interesting finding but is difficult to assess based the quantifications presented in Supp Fig 5. This claim could be strengthened by directly quantifying and comparing the distances between cristae bodies across variants to test whether I-OPA1 may be playing a role in maintaining cristae stacking/ordering.

The stacking phenotype was observed in all cells lines, but was more prevalent in I-Opa1* cells. Per the reviewer's suggestion, we attempted to use a recently published toolkit (Barad et al. 2023, PMID: 36786771) to quantify the intercristae distances, and evaluate if there is

uniformity in the distances between cristae in the cell lines. We purposefully selected mitochondria with stacking cristae from each cell line. The preliminary results of the toolkit are inconclusive as not all cristae in mitochondria classified into this category were stacking. The morphometrics toolkit takes segmented and labeled membranes, triangulates them and measures distances between specified nearest neighbors – in our case distances between cristae. These distances are measured throughout the entire 3D volume of the tomogram (or mitochondria) and can be presented as a histogram, resulting in a distribution of distances. Below are representative mitochondria from each cell line with distances between cristae plotted based by color (purple=0 and yellow=400nm) with histograms of distances on the right.

No distinct distribution patterns are observed in each cell line. In WT cells, cristae are shorter and thus when 3 cristae are considered, the measured distance from one crista to another may skip over a crista that ends earlier than the two surrounding it, leading to a wider range of inter-cristae distances. In s-Opa1* and Opa1-KO cells, there should be less uniformity to the inter-cristae distance, but this is hard to capture as some cristae are stacked.

We attempted to measure distances between CJs in order to bypass the problem of short cristae in WT mitochondria. These results are also inconclusive because not all CJs are captured within the lamellas.

Overall, the prevalence of this phenotype in l-Opa1* cells is striking and a qualitative observation. This phenotype can be the result of incomplete or delayed fusion because of the absence of s-Opa1 in these mitochondria, which would not be reflected in uniform distances between cristae across different mitochondria.

12. Apoptotic Assay: In line 321: "However, the importance of rearrangement and disassembly of Opa1 complexes during apoptosis initiation, to facilitate crista junction opening, as discussed above, demonstrates that Opa1 also has a direct role in facilitating the release of cytochrome c release when apoptosis is initiated. This view is consistent with our finding that complete knock-out of Opa1 impairs cytochrome c release and can be protective against apoptosis-inducing agents." I still don't quite follow the logic in this. How would wider junctions lead to higher apoptotic resistance? If we are following the model that cristae junction tightening is inhibits cyt c release, this seems counterintuitive. Perhaps incorporating these aspects into the model would help (see below).

Reviewer 3 correctly highlights a novel aspect of our findings (also noted by Reviewer 1) that s-Opa1* and Opa1-KO cells are resistant to apoptotic initiation, while their steady state cristae junction widths are wider than WT. This ultrastructure-function observation would indicate that it is not simply cristae junction width that is important in cytochrome C release. We do not speculate as to what the difference in widths may reflect. Even WT cristae junction widths are wide enough for a soluble factor to leak, and a simple diffusion barrier model may not be sufficient to explain retention of cytochrome C in the cristae lumen. Thus, cristae junction width may not be the only regulator for cytochrome C release. The observation of increased widths could be due to a change in cristae junction integrity or maintenance by l-Opa1, as our results suggest that l-Opa1 is required for WT-like response to apoptosis. Understanding the intermediates in the membrane transition resulting in

cytochrome c release are outside the scope of this work. We have adjusted our text to better convey these surprising results and their implications.

13. Model: Overall, the model presented is overly simplistic and does not adequately illustrate many of the exciting findings that were presented throughout the paper. As presented, the model does not substantially change the current view in the field, which I think is contradictory to what their data demonstrates, and I feel that the authors are selling themselves a bit short on their model. For example, it is missing any description of the findings related to: cristae volume, potential interaction of s-*opa1* at junctions, and the importance of l-*opa1** in mediating cristae stacking, etc. Additionally, the model is lacking any context relative to the defects observed in fission-fusion dynamics, and how these may be additionally coupled to differences in membrane ultrastructure described throughout the manuscript. The model also shows that s-*Opa1* plays a role in sustaining narrow tubular cristae, however Figure 2 demonstrates that the majority of cristae in S-*Opa1** exhibit lamellar cristae. Finally, one of the most exciting aspects of this paper is the attempt to link differences in cristae ultrastructure from OPA1 variants (i.e., form) to apoptotic priming (i.e., function). This aspect is not entirely explained within the model which, as presented, weakens the overall significance and impact of this work. Even if the authors are not entirely confident in the functional implications of the differences observed, it is worth attempting to illustrate their findings more effectively (and noting any uncertainties) to better convey how this study advances the field.

The Reviewer's statement that we are understating our results is encouraging. We have adjusted our model and discussion to reflect our exciting new results (Fig. 6).

Minor Concerns:

1. For a more general audience, it would be useful to include a more detailed rationale as to why cryo-ET may provide more insight to the role of OPA1 in mediating cristae structure compared to previous studies using fixed or heavy metal-stained TEM methods.

We thank the reviewer for their suggestions and have edited the text (Lines 5-9, 54-70, 303-310). Reviewer 1 also mentioned this.

2. How are the confidence intervals used when describing areas in figure 1c calculated? E.g. 0.34 {plus minus} 0.03 square microns for WT? The precision seems considerably lower when looking at the actual violin plot.

Regarding the violin plots, the bold line indicates median \pm quartiles and not mean \pm SD, therefore, to avoid confusion and provide a better representation of the raw data, we have changed all our plots from violin to scatter plots (see answer to comment 2 above).

3. For cristae density: what criteria was used to determine what constituted an individual crista?

Individual crista were identified based on previous literature (*Harner et al. 2016, PMID PMID: 27849155*). They are partitions of mitochondria formed by the infolding of IM. In our tomograms, invaginating membranes mainly formed continuous sheets or tubes along the 3D volume varying in width and length that enclose a lumen that follow. These invaginating membranes may or may not have an observable connection to the flat regions of the inner membrane of mitochondria.

4. Starting in line 159: "Indeed, cells with altered levels and forms of Opa1 (*Opa1*-OE, l-*Opa1**

and s-Opa1* lines) have a reduced proportion of tubular and globular cristae, with a greater variation of cristae shape observed in s-Opa1* cells." It appears wild type has the greatest variation (and not s-Opa1*)? Could the authors clarify this point?

We mean that s-Opa1* mitochondria have a greater variation of cristae shape compared to the other two before stated (Opa1-OE and I-Opa1*) Opa1 variants.

To avoid confusion, we have revised the description of cristae in lines 158-162:

'Interestingly, the proportion of lamellar cristae increases in I-Opa1* (81%) and s-Opa1* (77%) cells. Tubular cristae are not observed in I-Opa1* and Opa1-KO conditions and are reduced in Opa1-OE (3.24%) and s-Opa1* (2.3%) conditions compared to WT (9.6%). Globular cristae are present in all cell lines albeit to a lesser extent when Opa1 expression levels are altered but increased in the absence of Opa1 (Fig. 2b).'

5.It would be useful to include a few more sentences in the beginning of the BH3 profiling section to provide a smoother transition and rationale between cristae shape sections and functional assays.

We have added more functional data and attempted to smoothen the transition between cryo-ET and mitochondrial function sections. See transition starting at line 202.

6.Figure S2 appears to be missing panels (d,e).

We thank the reviewer for pointing out this error and have adjusted the legend.

7.The meaning of the ****'s differs from figure to figure which is quite confusing to read.

The reviewer makes an excellent suggestion. We have edited the figures to have consistent ** values.

8.Some of the classification was done by visual inspection as stated in the methods (1b,2b). If the authors could provide some information on how this classification was done. Was it an individual person who classified or were there a few people who classified based off of exemplar images?

These classifications were done by an individual person after putting together a gallery of exemplar images for group discussion, and all group lab members collectively agreed on classification.

9.In figure 2g the color rendering of the averages should correspond to the colors of the histograms (f) and quantifications (c,d).

We have fixed this in our images. Thank you for this suggestion.

10.How was targeting performed during tilt series acquisition? Was there bias in the targeting towards mitochondria, or mitochondria of a certain phenotype?

Targeting of mitochondria was done by two users. The typical guidelines were a double membrane architecture with visible invaginations of the inner membrane.

11.Starting in line 121: "Aligned with previous reports (Gómez-Valadés et al, 2021),

mitochondria from Opa1-KO cells are larger than WT, but have a similar abundance and total mitochondrial area per cell as WT (Fig. S4d)." This is a bit confusing and seems contradictory. In the light microscopy data (Supp Figure 3), wild type has extensive elongation and Opa1-KO shows highly fragmented mitochondria. Could the authors clarify this point more clearly?

We have edited accordingly to avoid confusing statements in the revised version.

12.Despite the wild type and I-Opa1* cells showing hyperfused mitochondrial networks (Supp Figure 3), the mitochondria imaged by conventional TEM in Supp Figure 4 are highly fragmented. Could the authors explain the discrepancy between mitochondrial morphology observed between these data?

For further clarity, we have omitted the TEM section of our data in this resubmission. Our fluorescent microscopy data and quantification correlate with our reported cryo-ET results and conclusions. We believe the TEM data are of good quality but their limitations (artifacts, covering only a section of a cell, limited to two dimensions), distract from our effort to succinctly highlight our cryo-ET data. Further studies comparing TEM and cryo-ET data are out of the scope of this work.

13.Starting in line 118: "Mitochondrial matrix density was classified as normal, dark, uneven or empty, which reflects brighter matrix staining (Fig. S4c). Matrix staining is prone to artifacts introduced by heterogenous heavy-metal stain and/or resin embedding." Could the authors explain the rationale in assaying the matrix texture using conventional TEM considering the described artifacts?

Please see answer above.

Non-essential suggestions for improving the study (which will be at the author's/editor's discretion)

1.Starting in line 326: "Mitochondrial shape and the curvature of the outer membrane has been previously found to affect the ability of the pro-apoptotic, pore-forming protein BAX to be stabilized in the mitochondrial outer membrane and initiate mitochondrial outer membrane permeabilization (MOMP) in cytochrome c release." Given the importance of curvature in mediating assembly of these essential pro-apoptotic proteins and the role of Opa-1-mediated apoptotic membrane remodeling, it may be quite interesting to examine how membrane curvature changes in the various Opa1 variants.

As we did not intend to make any claims on curvature in the manuscript, we have removed the word 'curvature' here to avoid misleading conclusions.

Dear Luke,

Thank you for submitting the revised version of your manuscript, which addresses the concerns of the referees. This revised version has now been re-reviewed; I attach the second referee reports to the bottom of this mail. The three reports were particularly diverse. While Referee 1 had significant concerns over a lack of conceptual advance, Referee 3 was fully supportive of publication due, primarily, to the technical accomplishment of your work. Referee 2 acknowledges both positions. After discussing the matter with my colleagues in the EMBO Journal editorial team, I have decided to proceed with your manuscript towards publication. Please prepare a point-by-point response to the referees' concerns. I encourage you, whilst taking the referees' comments on board, to be particularly careful to discuss those aspects of the current scientific discussion upon which your data most notably impinge.

Before I can formally accept your manuscript for publication, however, there are some remaining editorial points which need to be addressed. In this regard would you please:

- if appropriate, refer to any data repositories at which raw data are stored in a Data Availability Section,
 - rename the Conflict of Interest statement the 'Disclosure and Competing Interests' statement,
 - remove the author credit section from the manuscript,
 - include call-outs in the main text for Sub-figures 2G, 4D and 5B,
 - either indicate in figure legends wherever the same representative images are used in different panels (Figure 1A WT / Opa1OE vs Figure 3C WT / Opa1OE; Figure 1A vs appendix Fig S2A; Fig S4C and Fig S6B; Figure 3C vs appendix Fig S4C) or choose distinct representative images,
 - supply Source Data for the western blots in Fig. S1a
 - note that The EMBO Journal now uses a run-on style for figure legends, and amend appropriately,
 - consolidate data descriptions at the end of each figure legend into a separate 'Data Information' section,
 - in figures 1c; 4a; 5b; supplementary figures 2b; 10d correct the mismatches between the annotated p values in the figure legend and the annotated p values in the figure file,
 - define the annotated p values ****/**/**/* in the legends of figure 4b; supplementary figure 4a as appropriate,
 - indicate the statistical test used for data analysis in the legends of figures 4b and supplementary figure 4a,
 - define error bars in the legend of figures 4a, b, d; 5a-b; supplementary figures 9a-b; 10c-d; and 11d,
 - there are 11 Appendix figures (S1-S11); rename 5 of these to Figure EV1-EV5 and put their legends in ms file, with the corresponding callouts; the others should be compiled in and Appendix PDF with a table of contents with page numbers,
 - remove the legends of appendix figures from the manuscript file and place below each figure using the nomenclature Appendix Figure S1-S6 with the appropriate callouts in the text,
 - rename the six movie files Movie EV1-EV6 with the corresponding callouts, and remove their legends from manuscript files, zipping with each movie file,
- either leave Supplementary Table 1 in the ms file and rename to Table 1 with the corresponding callout - the separately uploaded table should then be removed; or remove from the ms file and rename to Table EV1 with the appropriate callout. It can also be included in Appendix PDF, renamed to Appendix Table S1 with the corresponding callout,
- correct the section order as follows: title page with complete author information, abstract, introduction, results, discussion, materials & methods, data availability section, acknowledgements, disclosure and competing interests statement, references, main figure legends, tables, expanded figure legends, and
 - check co-author email Leillani L. Ha - LEHA@mgh.harvard.edu

We include a synopsis of the paper (see <http://emboj.embopress.org/>). Please provide me with a two sentence general summary statement and 3-5 bullet points that capture the key findings of the paper.

We also need a summary figure for the synopsis. The size should be 550 wide by [200-400] high (pixels). You can also use something from the figures if that is easier.

EMBO Press is an editorially independent publishing platform for the development of EMBO scientific publications.

Best wishes,

William

William Teale, PhD
Editor
The EMBO Journal
w.teale@embojournal.org

We realize that it is difficult to revise to a specific deadline. In the interest of protecting the conceptual advance provided by the work, we recommend a revision within 3 months (14th Feb 2024). Please discuss the revision progress ahead of this time with the editor if you require more time to complete the revisions. Use the link below to submit your revision:

Referee #1:

When revising the manuscript, the authors have improved the wording and description of their findings, improved the statistical analysis and performed functional experiments, correlating calcium metabolism and respiratory activity with cristae structures in the different cell lines. While I generally appreciate the authors' efforts, I am afraid my concerns still hold. I had three main concerns: 1) limited novelty; 2) incomplete analysis of findings that appear to contradict previous reports; 3) use of the OMA1KO cells to allow accumulation of L-OPA1. There is no doubt that the analysis of the role of different OPA1 forms in cristae formation by cryo-FIB is of interest. I completely agree with the authors and do recognise the technical advance of cryo-FIB, as emphasised by the authors in their rebuttal letter. I also appreciate that improved discussion of apparent discrepancies with previous findings. However, in my opinion the manuscript still provides very little, if any, new mechanistic insight into the role of OPA1 forms in cristae formation and remains largely correlative, despite the additional functional experiments. The studies on calcium buffering capacity, respiratory function and mtDNA distribution show defects that are expected (and therefore were selected by the authors), but unfortunately provide little insight into the role of OPA1 or its different forms. The authors have analyzed the calcium buffering capacity and the respiratory function in OMA1KO cells and compared it with L-Opa1* (not mentioned in the main text). OMA1 deletion affected respiration, which raises questions about the (stronger) effect on respiration in L-Opa1* cells and which is also in apparent contradiction to several previous reports analyzing respiration in OMA1KO cells (e.g. Quirós et al., 2012). According to Figure S10, expression of Opa1V5 restores respiration in OMA1KO cells, which is difficult to reconcile with the role of OMA1 for OPA1 processing.

Referee #2:

The manuscript is much improved, although I still do not see a mechanistic study to explain the effects on Cyt C release or on PTP sensitivity. These are all interesting correlations. Yet, I do find these correlations valuable and exciting, and I also find the disagreement with previous studies interesting and valuable, as long as they are clearly pointed out and provide a potential explanation for the discrepancy.

A key minor issue is the lack of proper labeling of the Y axis of the graphs in multiple figures. Although this is not a major issue, it makes the difference between a paper that is pleasant to read and a tiring paper. The Y axis should report on A) the phenomenon of interest (example: branching), B) the parameter measured (example: Aspect ratio), and C) units. The words "ratio", "Percentage," or "relative fluorescence intensity" are not sufficient. All 3 categories have to appear in the Y axis. Sometimes the category of phenomenon can appear above the graph.

Referee #3:

Summary and Significance:

The authors present a technical tour-de-force manuscript addressing the important biological question, "how do different variants of OPA1 influence mitochondrial membrane ultrastructure and function?". The authors use state-of-the-art cellular cryo-electron tomography and advanced computational approaches to quantitatively compare mitochondrial membrane ultrastructures across cell lines expressing different variants of OPA1. The revised manuscript incorporates additional functional data to link these cristae ultrastructures to mitochondrial calcium and respiration. The authors also included additional analyses and details regarding their data processing and statistical methods. In its current form, the manuscript draws much stronger conclusions regarding the role of different OPA1 variants in mediating mitochondrial form and function. I anticipate it will be of broad interest to both the mitochondrial biology and cellular tomography fields.

Concerns:

The authors have addressed all my major concerns, including more details and rationale regarding their use of various statistical methods, data processing, and quantifications. Furthermore, the authors present a more robust, more convincing model that synergizes their findings. I have no further concerns about this manuscript. I support its acceptance for publication.

Referee #1:

When revising the manuscript, the authors have improved the wording and description of their findings, improved the statistical analysis and performed functional experiments, correlating calcium metabolism and respiratory activity with cristae structures in the different cell lines. While I generally appreciate the authors' efforts, I am afraid my concerns still hold. I had three main concerns: 1) limited novelty;

We acknowledge and respect the concerns raised by Referee #1. We believe the novelty of our work lies in systematic characterization of membrane morphologies in cell lines that differ in Opa1 processing. Relating new views and quantifications to functional differences should be considered as an important first step in understanding how Opa1 regulates the cristae membranes.

2) incomplete analysis of findings that appear to contradict previous reports;

In our revised manuscript, we take care to discuss the implications of the apparent contradiction with previous apoptosis finding (lines 336-346), while making sure to not overstep and overinterpret mechanisms, which will require follow-up study.

3) use of the OMA1KO cells to allow accumulation of L-OPA1.

Throughout the text we take care to discuss the differences in OMA1-KO and l-Opa1* cells:

- a. With regard to BH3 profiling: “Like l-Opa1* cells, *omal*^{-/-} cells also have WT levels of cytochrome c release upon treatment with BIM” (lines 268-269). Thus, neither the *omal*^{-/-} background nor the accumulation of l-Opa1* affects the cell response to apoptotic stimulants”.
- b. With regard to calcium handling: “Additionally, *omal*^{-/-} MEF cells show WT-like propensity for mPTP opening under Ca²⁺ stimulation” (lines 269-270). Again, neither the *omal*^{-/-} background nor the accumulation of l-Opa1* affects calcium handling.
- c. With regard to respiration: “OCR values in l-Opa1* cells are significantly lower than in *omal*^{-/-} cells, indicating the *OMA1* deletion is not the sole contributor to the impaired respiration observed in l-Opa1* cells”. (lines 271-273) Thus, the imbalance favoring the l-Opa1 form compounds on the impairment of respiration.

Additionally, in many cases, deviations from WT behavior (i.e. BH3 profiling and calcium handling) were observed in the s-Opa1* cell lines, which were generated with a *omal*^{+/+} background.

There is no doubt that the analysis of the role of different OPA1 forms in cristae formation by cryo-FIB is of interest. I completely agree with the authors and do recognise the technical advance of cryo-FIB, as emphasised by the authors in their rebuttal letter. I also appreciate that improved discussion of apparent discrepancies with previous findings. However, in my opinion the manuscript still provides very little, if any, new mechanistic insight into the role of OPA1 forms in cristae formation and remains largely correlative, despite the additional functional experiments. The studies on calcium buffering capacity, respiratory function and mtDNA distribution show

defects that are expected (and therefore were selected by the authors), but unfortunately provide little insight into the role of OPA1 or its different forms.

Although the referee deems our results to be expected, we believe it is important to conduct these experiments to test and validate expectations. The outcome of these experiments also allowed us to correlate specific cristae morphologies to mitochondrial dysfunctions (See Fig. 6). Specifically, wider cristae junctions and vesicular cristae were observed to correlate with abnormal apoptotic priming and calcium handling, as well as amorphous cristae correlated with compromised mitochondrial respiration. We would argue that one unexpected finding was that s-OPA1* and KO-OPA1 cells showed resistance to apoptosis, even though in steady-state their CJs were wider than CJs in WT cells. Previous work demonstrated that CJ widening occurs in response to apoptosis, but our work suggests that this apoptosis induced widening of CJs is independent of the steady-state width.

The authors have analyzed the calcium buffering capacity and the respiratory function in OMA1KO cells and compared it with L-OPA1* (not mentioned in the main text).

As we feel the efforts we made to distinguish functional impairments due to the imbalance of Opa1 and not the presence or absence of Oma1 are important for our manuscript, we did reference this figure (Appendix FigS5) in the text in lines 267-273):

“Mitochondrial fitness was also measured in the *oma1*^{-/-} background in which the l-OPA1* cells were generated (Wang *et al*, 2021). Like l-OPA1* cells, *oma1*^{-/-} cells also have WT levels of cytochrome c release upon treatment with BIM (**Appendix Fig S5A**). Additionally, *oma1*^{-/-} MEF cells show WT-like propensity for mPTP opening under Ca²⁺ stimulation albeit displaying a lower overall calcium buffering profile than WT and l-OPA1* cells (**Appendix Fig S5B**). OCR values in l-OPA1* cells are significantly lower than in *oma1*^{-/-} cells, indicating the *OMA1* deletion is not the sole contributor to the impaired respiration observed in l-OPA1* cells (**Appendix Fig S5C and D**). These data suggest that functional differences in the l-OPA1* mitochondria are dependent on the form of Opa1 present.”

OMA1 deletion affected respiration, which raises questions about the (stronger) effect on respiration in L-OPA1* cells and which is also in apparent contradiction to several previous reports analyzing respiration in OMA1KO cells (e.g. Quirós *et al.*, 2012).

The stronger effect on respiration in l-OPA1* cells demonstrates that the Opa1 imbalance also affects respiration. We do not agree that our work contradicts previous work. In reference to the call out by the referee, Quiros and colleagues do report impaired respiration in Oma1-KO cells. The role of Opa1 and Oma1 in brown adipocyte function is explored in this study, but the authors introduce a mutant form of Opa1 that cannot be cleaved by Oma1. Oma1 is not the only IMM protease that can cleave Opa1, Opa1 also possesses a Yme1L cleavage site. We address this in our work by removing a Yme1L cleavage site (Δ *exon5b*) in our l-OPA1* cells.

According to Figure S10, expression of Opa1V5 restores respiration in OMA1KO cells, which is difficult to reconcile with the role of OMA1 for OPA1 processing.

We do not express Opa1-isoform5 (nor Opa1V5) in a *oma1*^{-/-} background. The s-Opa1* cells were generated by expressing Opa1-isoform5 in a *opa1*^{-/-} background. Fig 5a demonstrates that this cell line (s-Opa1*) also had impaired mitochondrial respiration. In FigS10, we compare WT, l-Opa1*, and *oma1*^{-/-} cells. If we were to add our s-Opa1* cells into this plot (although this cell line is in an *oma1*^{+/+} background) we see that s-Opa1* and *oma1*^{-/-} have similar levels and does not restore respiration:

Referee #2:

The manuscript is much improved, although I still do not see a mechanistic study to explain the effects on Cyt C release or on PTP sensitivity. These are all interesting correlations. Yet, I do find these correlations valuable and exciting, and I also find the disagreement with previous studies interesting and valuable, as long as they are clearly pointed out and provide a potential explanation for the discrepancy.

We thank Referee #2 for acknowledging the value and exciting nature of our observations. To provide context to our disagreement with previous studies, we have taken care to outline and caveat the limitations of our MEF model system. In particular, we make efforts to point out differences between previous work and our findings in the discussion (lines: 318-342):

“In contrast to previous studies, we investigated cristae state prior to any exposure to extracellular stress or apoptotic stimuli (Merkwirth *et al*, 2008, 2012). Building upon previous studies, we observed wider cristae junctions in Opa1-KO and s-Opa1* cells prior to apoptotic initiation (**Fig. 3**)...Our findings that Opa1-KO cells are resistant to apoptosis may seem counterintuitive given previous reports of apoptotic resistance in Opa1-overexpressing cells (Frezza *et al*, 2006). However, this apparent discrepancy can be reconciled if considering multiple roles for Opa1 during apoptotic membrane transitions. During apoptosis, Opa1 maintains cristae junctions, which may restrict cytochrome c release in response to pro-apoptotic signals (Frezza *et al*, 2006). However, the importance of rearrangement and disassembly of Opa1 complexes during apoptosis initiation, to facilitate CJ opening, as discussed above, demonstrates that Opa1 also has a direct role in the earlier steps when

apoptosis is being initiated (**Fig. 6d**). This view is consistent with our finding that complete knock-out of Opal impairs cytochrome c release and can be protective against apoptosis-inducing agents (**Fig. 4a-b**). Our data indicate that apoptotic response is independent of Opal processing (Cipolat *et al*, 2006) and cytochrome c release is dependent on initial CJ widths.

Our BH3 profiling results are supported by CRC assay results, where l-Opal presence in MEF allows for WT-like responses towards increased Ca^{2+} and apoptotic stimulation (**Fig. 4c-d**). These independent functional outcomes are connected by the importance of membrane integrity in mPTP opening (Strubbe-Rivera *et al*, 2021; Bernardi *et al*, 2023) and BAX/BAK-mediated cytochrome c cristae remodeling (Scorrano *et al*, 2002; Renault *et al*, 2015). Consistent with previous reports of calcium phosphate deposits, we observed dense deposits in the mitochondria matrix of most Opal-OE mitochondria (Wolf *et al*, 2017), which correlates with a lower mPTP transition CaCl_2 threshold.”

A key minor issue is the lack of proper labeling of the Y axis of the graphs in multiple figures. Although this is not a major issue, it makes the difference between a paper that is pleasant to read and a tiring paper. The Y axis should report on A) the phenomenon of interest (example: branching), B) the parameter measured (example: Aspect ratio), and C) units. The words "ratio", "Percentage," or "relative fluorescence intensity" are not sufficient. All 3 categories have to appear in the Y axis. Sometimes the category of phenomenon can appear above the graph.

We thank the referee for this suggestion and have corrected the labeling of the y-axis throughout.

Referee #3:

Summary and Significance:

The authors present a technical tour-de-force manuscript addressing the important biological question, "how do different variants of OPA1 influence mitochondrial membrane ultrastructure and function?". The authors use state-of-the-art cellular cryo-electron tomography and advanced computational approaches to quantitatively compare mitochondrial membrane ultrastructures across cell lines expressing different variants of OPA1. The revised manuscript incorporates additional functional data to link these cristae ultrastructures to mitochondrial calcium and respiration. The authors also included additional analyses and details regarding their data processing and statistical methods. In its current form, the manuscript draws much stronger conclusions regarding the role of different OPA1 variants in mediating mitochondrial form and function. I anticipate it will be of broad interest to both the mitochondrial biology and cellular tomography fields.

Concerns:

The authors have addressed all my major concerns, including more details and rationale regarding their use of various statistical methods, data processing, and quantifications. Furthermore, the authors present a more robust, more convincing model that synergizes their findings. I have no further concerns about this manuscript. I support its acceptance for publication.

We thank Referee #3 for acknowledging the impact and conclusions of our work and their kind words.

Dear Luke,

I am pleased to inform you that your manuscript has been accepted for publication in the EMBO Journal.

Congratulations! I am delighted to see this article in The EMBO Journal.

Best wishes,

William

William Teale, PhD
Editor
The EMBO Journal
w.teale@embojournal.org
